# Calculation of the Isobaric Heat Capacities of the Liquid and Solid Phase of Organic Compounds at and around 298.15 K Based on Their “True” Molecular Volume

**DOI:** 10.3390/molecules24081626

**Published:** 2019-04-24

**Authors:** Rudolf Naef

**Affiliations:** Department of Chemistry, University of Basel, 4003 Basel, Switzerland; rudolf.naef@unibas.ch; Tel.: +41-61-9119273

**Keywords:** heat capacity, molecular volume, force-field geometry optimization, hydrocarbons, ionic liquids, siloxanes, metal complexes

## Abstract

A universally applicable method for the prediction of the isobaric heat capacities of the liquid and solid phase of molecules at 298.15 K is presented, derived from their “true” volume. The molecules’ “true” volume in A^3^ is calculated on the basis of their geometry-optimized structure and the Van-der-Waals radii of their constituting atoms by means of a fast numerical algorithm. Good linear correlations of the “true” volume of a large number of compounds encompassing all classes and sizes with their experimental liquid and solid heat capacities over a large range have been found, although noticeably distorted by intermolecular hydrogen-bond effects. To account for these effects, the total amount of 1303 compounds with known experimental liquid heat capacities has been subdivided into three subsets consisting of 1102 hydroxy-group-free compounds, 164 monoalcohols/monoacids, and 36 polyalcohols/polyacids. The standard deviations for Cp(liq,298) were 20.7 J/mol/K for the OH-free compunds, 22.91 J/mol/K for the monoalcohols/monoacids and 16.03 J/mol/K for the polyols/polyacids. Analogously, 797 compounds with known solid heat capacities have been separated into a subset of 555 OH-free compounds, 123 monoalcohols/monoacids and 119 polyols/polyacids. The standard deviations for Cp(sol,298) were calculated to 23.14 J/mol/K for the first, 21.62 J/mol/K for the second, and 19.75 J/mol/K for the last subset. A discussion of structural and intermolecular effects influencing the heat capacities as well as of some special classes, in particular hydrocarbons, ionic liquids, siloxanes and metallocenes, has been given. In addition, the present method has successfully been extended to enable the prediction of the temperature dependence of the solid and liquid heat capacities in the range between 250 and 350 K.

## 1. Introduction

The heat capacity is a fundamental extensive property characterizing a molecule’s thermophysical response towards the addition of heat energy, knowledge of which is important e.g., in connection with the adjustment calculation of further thermodynamic properties such as the heats and entropies of sublimation, vaporization or solvation to a defined temperature. The theoretical approach for the evaluation of the heat capacity Cp at constant pressure and at a given physical phase state and temperature is feasible for single atoms and basically only for very small molecules due to the rapid increase of the degrees of freedom of translational, rotational, vibrational and electronic motion with the number of atoms in a molecule and the question as to which of these degrees of freedom are excited at all at a certain temperature. Based on these findings, it is clear that the heat capacity of a molecule is temperature-dependent, being zero at 0K and increasing with temperature. As for larger molecules, various experimental calorimetric methods have shown to provide an easy experimental access to their values. Among these, the most popular instruments are the adiabatic calorimetry (AC) [1], the differential scanning calorimetry (DSC) [2], and the modulated DSC (MDSC) [3,4]. These methods have successfully been applied to all kinds of molecules besides ordinary organic compounds: literature references of experimental data for inorganic and organic salts, liquid crystals, ionic liquids (ILs) as well as metal-organics have been found and will be cited later on. 

Based on these experimental data, a large number of methods for the prediction of the heat capacities of as yet unknown compounds have been developed, the most prominent of which are founded on the group-additivity (GA) approach [5,6,7,8,9,10,11,12,13,14,15,16,17,18]. Chickos et al. [5] described a GA method for the prediction of the heat capacities of 810 liquids and 446 solids using 47 functional groups, which take account of the carbon atoms’ substitution and hybridization states, yielding a standard error of 19.5 J/mol/K for liquids and of 26.9 J/mol/K for solids. They compared these standard deviations with the experimental standard errors of 8.12 and 23.4 J/mol/K respectively, derived from the numerous experimental data variations for 219 liquids and 102 solids cited by several independent sources. Ruzicka and Domalski [6] developed a second-order GA method for the prediction of the heat capacity of 265 liquid hydrocarbons as a function of temperature between their melting and normal boiling point, introducing a polynomial expression, taking into account the temperature-dependence of the group parameters. The mean percentage deviation (MPD) of their prediction was given as 1.9%. The same authors extended their GA method to liquid compounds carrying halogens, nitrogen, oxygen and sulfur [7], enabling the estimation of the heat capacity of a total of 558 liquids with an MPD of 2.9%. Later on, Zàbransky and Ruzicka [8] presented an amended GA method based on the one published by Ruzicka and Domalski [6,7], which extended the number of functional groups to 130, including *cis*, *trans* as well as *ortho* and *meta* corrections in the GA parameters calculations, reporting deviation errors of between 1 and 2%. A setback was the observation that the results deteriorated if the compound contained functional groups from more than one family, as e.g., in *N*,*N*’-diethanolamine or 1-chloro-2-propanol. Another approach was chosen by Goodman et al. [9] in that they tested two different equations to account for the temperature dependence of the heat capacity C_p_, the simple power-law form C_p_ = AT^m^, where A and m are empirical coefficients and m is less than 1, and a more elaborate form based on the Einstein-Debye partition function for crystals applying a modified frequency-distribution function. The MPD for 455 compounds in the training set amounted to 6.8% for the former formula and 8% for the latter function. A novel three-level GA method was introduced by Kukal et al. [10], the levels being defined as increasingly complex molecule fragments. In addition, the temperature dependence of the heat capacity in the range between the melting and the normal boiling point was included in a polynomial formula for the contributions of these fragments. Based on 549 compounds, an MPD of 1.2% over the complete set over the mentioned temperature range was found. At 298.15K, an MPD of 1.5% resulted for 404 compounds of the basis set and 2.5% for 149 compounds for an independent test set. In recent years the GA approach has found particular interest in the estimation of the heat capacity of ILs [11,12,13,14,15,16]. Gardas and Coutinho [11] presented the results of the predictions of the heat capacity and its temperature relation for a series of 19 ILs consisting of imidazolium, pyridinium and pyrrolidinium cations and several mono- and polyatomic anions, based on the second-order GA method described by Ruzicka and Domalski [6,7]. The groups therein represented the complete anions and the cations minus the alkyl chains, while their methylene and methyl functions were treated separately, which limits the method’s applicability to just this kind of ILs. Accordingly, the Cp predictions deviated for 90.2% by less than 1% from the experimental data. They also observed an interesting correlation of the heat capacities with the ILs’ molar and molecular volumes, which will be the subject of further discussion. Ceriani et al. [12] used a GA method for the prediction of the heat-capacities and their temperature dependence of 1395 fatty compounds contained in edible oils and biofuels, encompassing acids, alcohols, esters, triacylglycerols as well as hydrocarbons, applying only seven functional groups, whereby each group contribution was represented by a temperature-dependent linear function. They reported an MPD of only 2.6%. In contrast to Gardas and Coutinho’s [11] paper, Valderrama et al. [13] refined the functional groups constituting a set of 32 ionic liquids into 42 smaller fragments and added the heat-capacity data of 126 further ordinary compounds to the database in order to achieve a more general applicability. Beyond this, they added a term called mass connectivity index, introduced by Randic [14] and extended by Valderrama and Rojas [15], representing the degree of branching to the GA calculation. The MPD for the 32 ILs was calculated to 2.8%. In order to find the most effective functional groups in the GA method for the heat-capacity prediction of ILs, Sattari et al. [16] used a genetic function approach, which required the investigation of a series of models with increasing numbers of functional groups. They showed that a model with the 13 most effective groups produces sufficiently reliable results, yielding an overall correlation coefficient R^2^ of 0.99 and standard deviation error of 18.42 J/mol/K and an MPD of 1.68% for a total of 82 ILs. Albert and Müller [17] applied a GA model including a polynomial function with a degree of two for the temperature dependence of the heat capacities of ILs based on 36 functional groups and seven heterocyclic and heteroaromatic rings as corrective groups, yielding a standard deviation of 36 J/mol/K (MPD = 2.58%) for the training set of 86 ILs and 47 J/mol/K (MPD = 5.44%) for the test set of 20 ILs. A critical evaluation of the previous GA methods for the prediction of the heat capacities of ILs has been presented by Nancarrow et al. [18], whereby the methods have been distinguished into two approaches, the Meccano and the Lego approach. The former one describes a GA method wherein the functional groups consist of the complete building blocks of cations and ions, e.g., as presented by Gardas and Coutinho [11], whereas in the latter one the groups represent single atoms or small molecular fragments, exemplified by the method of Valderrama et al. [13]. Based on two sets of ILs for the respective GA methods they reported the obvious: while the Meccano approach was found to provide slightly lower deviations with experiment for the 45 test ILs, but was of limited applicability, the Lego approach, applied on 92 ILs, produced less accurate results but was more widely usable.

Besides the GA approach, a number of further methods for the prediction of the heat capacities have been used [19,20,21,22,23,24,25,26,27,28,29]. Morad et al. [19] estimated the heat capacities of fatty acids, triacylglycerols and vegetable oils by means of the Rowlinson-Bondi equation [20], which evaluates the difference between the liquid specific heat capacity and the ideal gas liquid specific heat capacity, based on the reduced temperature and an acentric factor. The heat capacity of the ideal gas again was calculated on the basis of the GA method of Rihany and Doraisamy [21]. An experimental approach based on the vibrational spectra and a direct calculation of the spectra using density functional theory was used for the prediction of the heat capacity of polynuclear aromatic solids by Sallamie and Shaw [22]. A method especially designed for the heat-capacity prediction of ILs was provided by Müller and Albert [23], in that they evaluated the contribution of each of 39 cations and 32 anions and their temperature dependence to the IL’s heat capacity. They reported a standard deviation of 13.2 J/mol/K for the training set and 30.1 J/mol/K for the test set. Barati-Harooni et al. [24] compared three Cp-prediction models, one called coupled simulated annealing least square support vector machine (CSA-LSSVM), a second one being a gene expression programming algorithm (GEP), and a third one called adaptive neuro fuzzy inference system (ANFIS). Applied on 56 ILs with 2940 data points, the results showed the best results with CSA-LSSVM, yielding a correlation coefficient R^2^ of 0.9933 and a standard deviation of 9.99 J/mol/K. An interesting approach was presented by Preiss et al. [25] in that they correlated the molecular volumes of ILs with their experimental heat capacities. They defined an IL’s molecular volume (in nm^3^) as the sum of the molecular volumes of its constituting cation and anion. Since these volumes have been derived from the unit cell dimensions of crystal structures obtained from x-ray measurements, this method’s scope of use was limited to the sets of ILs only consisting of the named anions and cations. Nevertheless, the fact that the results showed a linear relationship between the ILs’ volumes and their heat capacities over a Cp range of between ca. 300 and 1300 J/mol/K with a correlation coefficient R^2^ of ca. 0.97 and an MPD of ca. 4–5% (based on 34 ILs) not only confirmed the observation of Gardas et al. [11], but also indicated that the prediction of any molecules’ heat capacities via their molecular volumes should be feasible, on condition that a reliable and general method for the calculation of the molecular volumes is available. A similar linear correlation was found by Paulechka et al. [26] between the molar volumes (in m^3^/mol) and the heat capacities for 19 ILs, for which the authors calculated a standard deviation of 6.7 J/mol/K at 298.15K. Again, due to the requirement of the knowledge of the ILs’ density, the calculation of the heat capacity via the density would be limited. Later called “volume-based thermodynamics” (VBT) [27], Glasser and Jenkins [28] extended this approach to minerals and ionic solids and liquids, demonstrating the validity of Neumann–Kopp’s rule [29] of the additivity of the heat capacity contribution per atom.

All the presented methods for the prediction of the heat capacity of organic molecules, despite their usefulness and reliability within their designed range, have the disadvantage of not being generally applicable. In principle, the GA approach can solve this deficiency, e.g., following the GA approach of Ruzicka and Domalski [6,7], provided that the number of published experimental data is large and the molecules structurally versatile enough. Unfortunately, this is not the case. In the ongoing project ChemBrain, a recent extension of its versatile GA method for the prediction of a large number of molecular descriptors [30,31,32,33], now also enabling the prediction of the liquid and solid heat capacities (due for publication), has revealed that, due to the limited number of experimental data, the liquid heat capacity of only ca. 62% and the solid heat capacity of only ca. 65% of the compounds have been evaluable in its database of currently ca. 31’600, the structural diversity of which can be viewed as representative for the entire space of chemical structures. Therefore, a method was sought which preferably required only one molecular property, reliably computable for each and every molecule, from which its heat capacities could be derived. Gardas et al. [11] and Glasser and Jenkins [28] opened a possible solution in that the VBT approach refers to a property that is inherent in each molecule: its volume. However, although there are various experimental methods for the evaluation of a molecule’s volume, it would not make any sense to derive it from experimental data, e.g., x-ray or density, as these data are also limited in number. The present paper offers a reliable and reproducible way for the calculation of the “true” molecular volume, which only depends on the molecules’ 3-dimensional geometry and the Van-der-Waals (vdw) radii of their constituting atoms. It will demonstrate that these volumes excellently linearly correlate with the heat capacity of the solid and liquid phase of any class, size and structure of molecules, not only at standard conditions but also over a large temperature range. Thanks to the unlimited scope of its applicability, this study will also provide an awareness of the forces influencing the heat capacities, in particular the structural or intermolecular effects that are often camouflaged by other methods, and it will enable a reliable estimate of their magnitude. The limitless range of its applicability, on the other hand, forestalls a direct comparison of the quality of the present method with any of the results cited above, as none of these was based on a similarly comprehensive set of compounds, which encompasses molecules as diverse as alkanes, ionic liquids, siloxanes and metal complexes, to name a few. Therefore, the reliability of the present results has been put in relation to the scatter of the experimental heat capacities, the amount of which can best be assessed by the Cp values of several homologous series of compounds for which the increase between the consecutive elements should be approximately constant, examples of which will be presented.

## 2. General Procedure

The present method for the calculation of the molecular heat capacities is based on a database of currently ca. 31’600 molecules, each of them stored as geometry-optimized 3D structure in an object-oriented datafile, together with a number of experimental and routinely calculated thermodynamic and further descriptors, among which the experimental heat-capacity Cp(liq,298) data at 298.15K for 1303 liquids and the Cp(sol,298) data for 800 solids are to be mentioned in this context. In the following, the two decisive preconditions for a reliable heat-capacity prediction will be outlined in detail, the optimization of the molecular 3D geometry as the first step in the process, followed by the calculation of the “true” molecular volume. It will be shown in the results section that a third step is required, distinguishing between OH-group-containing and OH-free molecules, to enable the selection of the correct parameters for the linear equation leading to a reliable prediction of the heat capacities of their liquid and solid phases.

### 2.1. Geometry Optimization

The goal of the geometry optimization process is to ensure that the final bond lengths and angles in a molecule’s 3D presentation correspond to standard bond lengths and angles, as e.g., listed in the CRC Handbook of Chemistry and Physics [34], and that the intramolecular interactions of vdw radii [35] have been minimized. This optimization process in the ongoing project ChemBrain IXL is carried out by means of a force-field method called “steepest descent”, e.g., described by O. Ermer [36], meaning that each step of the iterative process leading to the energy minimum is controlled by the largest value of the first derivatives in x-, y- and z- direction of Equation (1) at each atom, evaluated by moving its position in each direction by a tiny amount and calculating the corresponding energy change.
(1)$$E=\sum E_{str}+\sum E_{ang}+\sum E_{tor}+\sum E_{vdw}+\sum E_{oop}$$

In this equation, the term E denotes the total energy which is defined as the sum of all bond stretching (*E_str_*), bond angle (*E_ang_*), torsional angle (*E_tor_*), vdw interaction (*E_vdw_*) and out-of-plane bending (*E_oop_*) energies. By default, during the optimization process the algorithm periodically scans the energy hyper-surface for the molecule structure’s global energy minimum, which e.g., results in the linearization of long-chain alkyl groups. As will be shown, however, achievement of the global energy minimum is not required, as the energy itself of the optimized structure is irrelevant for the present task.

### 2.2. Calculation of the “True” Molecular Volume

A number of publications [35,37,38,39,40,41] have been dealing with the calculation of the molecular volume. Several of them based the calculation on a physical property of the molecule such as the density, e.g., Kurtz and Sankin [37] and Ye and Shreeve [38], or the crystal structure, e.g., Jenkins et al. [39]. The disadvantage of these approaches lies in the fact that they require an examined physical property as a starting point. Beyond this, the evaluated volumes are not really “true” molecular volumes as they also include parts of the empty space around them. A purely analytical approach was chosen by Conolly [40], in that he constructed the molecular volume from a collection of spheres, consisting of partitions of the atomic vdw volumes and of solvent-excluded volumes. Although not dependent on any experimentally determined property, a computational algorithm based on this procedure seems far from easily generalizable. Gavezotti [41], in contrast, suggested a numerical method which starts from a molecule-enveloping space, containing a very large number of spatially randomly distributed probe points, counting all of those points inside any of the atomic vdw spheres and dividing this number by the total number of all probes. Multiplication of this fraction with the volume of the envelope space yielded the molecular volume. It will be evident that this approach fairly closely resembles the one which will now be outlined here.

The evaluation of the “true” molecular volume is carried out in the following way: at first, a rectangular box is defined, the sidelengths of which in x-, y- and z-direction are defined by the corresponding extensions of the molecule including the atomic vdw radii. Then a cube of a defined sidelength is systematically scanned through the box in all three directions in steps of the cube’s sidelength. Whenever the cube center is within the range of any of the atoms’ vdw radii, the cube’s volume is added to a container, which constitutes the “true” molecular volume. In order to speed up this procedure, the algorithm in each consecutive scan in x- and y-direction first sets up a list of those atoms whose vdw radii are within the cube’s reach in both directions, before scanning the third, i.e., the z-direction. In other words, the scan in the z-direction is reduced to those cubes that are within the molecule’s projection onto the xy-surface of the box. The molecular volume of salts with both anions and cations consisting of polyatomic structures, such as ILs, are calculated just like ordinary neutral molecules. However, in this case it is important to ensure that the two ions are well separated in order to prevent intermolecular overlap. As this procedure does not include monoatomic ions occurring in organic salts and ILs, because they are added separately to the formula, the scanning procedure in this case is completed by the addition of the ions’ individual vdw volume [42] to the container.

The cube’s sidelength has been defined in this project as 0.05 Angstroms (5 pm), resulting in a cube volume of 0.000125 A^3^ (125 pm^3^). Accordingly, the “true” molecular volume is given in A^3^. Despite the cube’s small size, the evaluation of the molecular volume of even a large molecule is carried out within a split second on a desktop computer. 

## 3. Results

### 3.1. General Remarks

Both the lists of molecules used in the correlation calculations of the heat capacities of the solid and liquid phase are collected in standard SDF files and are stored in the Appendix A, ready to be imported by external chemistry software. The Appendix A also provides the molecules lists used for each of the following correlation diagrams containing molecule names, experimental and prediction values. Additionally, it also contains the lists of outliers in the solid and liquid heat-capacity calculations.

### 3.2. “True” Molecular Volumes 

As mentioned in the prior section, the present procedure for the calculation of the molecular volume largely corresponds to the one of Gavezotti [41]. Correspondingly, one would expect similar results. This is indeed the case as is demonstrated on Table 1.

The values of the molecular volumes are not dependent on the geometry having the lowest global energy minimum, as is demonstrated in Figure 1, where tristearin, an unsaturated longchain glyceryl triester, is compared in a stretched and a randomly folded structural form. Their molecular volumes differ by less than 1%, the difference being owed to some intramolecular vdw overlaps.

The first tentative rounds of heat-capacity calculations revealed that for ionic liquids—when using standard bonds and vdw radii—the predicted heat capacities were generally too low by ca. 20 J/mol/K, i.e., by ca. one standard deviation. This deficiency was remedied by modifying the bond lengths of the bonds between the central, formally charge-carrying, atom in the ions and its neighbors, in that in the case of the anions the bonds have been extended by ca. 10% and in the cations they have been shortened by the same amount. In addition, for polyatomic anions the central atom’s vdw radius has been enlarged as suggested in [42] (for monoatomic anions the vdw values are given as listed in [42]).

Taking these modifications into account, the calculation of the molecular volumes of ILs as well as their constituting cations and anions is straightforward. In order to enable readers interested in the “true” molecular volumes of ILs and subsequently their heat capacities, Table 2 lists a number of anions and cations and their calculated molecular volume which proved to be the most popular in the heat-capacity studies. Thus, the “true” molecular volume of an IL consisting of any of the cations and anions of Table 2 is simply the sum of their partial volumes. Minor differences with the directly calculated molecular volumes of corresponding ILs have to be ascribed to slight differences in the optimized geometry as discussed above.

### 3.3. Sources of Heat-Capacity Data 

The majority of experimental heat-capacity data has been found in several collective papers, of which Zabransky et al. collected a series of liquid homologous linear alkanes [43] and 1-alkanols [44], while Costa et al. [45] complemented these series by further homologues of liquid linear 1-alkenes, 1-alkynes, 1-alkylthiols, 1-alkylamines, 1-nitro- and 1-haloalkanes. Domalski and Hearing [46] contributed heat-capacity data of liquid and solid, saturated and unsaturated, linear and cyclical hydrocarbons. The same authors provided a large compilation of further data of liquid and solid molecules possessing from one to 1077 carbons [47,48]. Another large collection of data for liquids and solids was provided in the section “Standard Thermodynamic Properties of Chemical Substances” of ref. [34]. Beyond these compilations a large number of recent papers presenting the heat-capacity data of several classes of compounds and of individual molecules of any kind as well as their temperature dependence have been published up to the present. Special mention shall be given to the hydrocarbons [49,50,51,52,53,54,55,56,57,58,59,60,61,62,63,64,65], halogenated hydrocarbons [66,67,68,69,70,71], unsubstituted and substituted alcohols and polyols including sugar derivatives [72,73,74,75,76,77,78,79,80,81,82,83,84,85,86,87,88,89,90,91,92,93,94,95,96,97,98,99,100,101,102,103,104,105,106,107], phenol derivatives [108,109], carboxylic acids [110,111,112,113,114,115,116,117,118], esters [119,120,121,122,123,124,125,126,127,128,129,130,131,132,133,134,135,136,137,138], anhydrides [139,140,141], aldehydes and ketones [142,143,144,145,146,147,148,149,150], ethers [151,152,153,154,155], amines [156,157,158,159,160,161,162], imines [163], oximes [164,165], anilines [166,167], amides [168,169,170,171], imides [172], barbiturates [173,174,175,176], ureas [177,178,179,180,181], hydantoins [182,183], carbamates [184,185], isocyanates [186], thiols, thio ethers and disulfides [187,188,189,190,191], sulfonamides [192,193,194], nitriles [195], metal complexes [196,197,198,199], silanes and siloxanes [200,201,202,203,204], nucleic bases and nucleosides [205,206,207], amino acids and peptides [208,209,210,211,212], unsubstituted and substituted hetarenes and heterocycles [213,214,215,216,217,218,219,220,221,222,223,224,225,226,227,228,229,230,231,232,233,234,235,236,237,238,239,240,241,242,243], high-energy nitro compounds [244,245] and various more [246,247,248,249,250,251,252,253,254,255,256,257,258,259,260]. 

In 2010, as the ILs gained increasing interest, Paulechka [261] sampled the room-temperature heat capacities of 102 ILs, covering the time from 1998 to 2010. In addition, a number of further IL data have been provided by the National Institute of Standards and Technology [262]. In the meantime, a number of recent papers have been published presenting further heat-capacity data of ILs [263,264,265,266,267,268,269,270,271,272,273,274,275,276,277,278,279,280,281,282,283,284,285,286,287].

### 3.4. Heat Capacity of Liquids

As mentioned in the introductory section, the VBT approach of Gardas et al. [11] and Glasser and Jenkins [28] pointed to a possible way to overcome the GA models’ disadvantage of not being globally applicable. Since ChemBrain’s database enables the search for correlations between any two molecular descriptors, it was obvious to try to find one—in accordance with the VBT approach—between the heat capacities and the “true” molecular volume, calculated as described in Section 3.2. The correlation between the “true” molecular volume and the experimentally measured liquid heat capacity of 1303 compounds of all classes indeed revealed a surprisingly good linearity over a large volume range, as shown in Figure 2, with a correlation coefficient of 0.9785 after removal of the worst outliers, despite a fairly large standard error of ca. 28 J/mol/K and a mean absolute percentage deviation (MAPD) of 8.23%.

However, a thorough analysis of the corresponding histogram (Figure 3) pointed to a severe deficiency of this preliminary correlation attempt, reflected in its apparent asymmetry with an overweight to the right side and a shift of the maximum to the left: for molecules carrying hydroxy groups, i.e., alcohols, carboxylic acids and strong acids, the predictions were systematically well below the experimentally determined heat capacities by up to ca. 130 J/mol/K! This deficiency evidently resulted from the neglect of the strong polarity of the OH group in alcohols and acids, apparently increasing the liquid heat capacity by the additional intermolecular O-H interaction, which is to be the subject of detailed discussion in Section 3.4.2.

#### 3.4.1. Heat Capacity of Liquids of Molecules Free of Hydroxy Groups 

The intermolecular hydrogen-bridge effect found in alcohols and acids evidently required a separation of these molecules from those without this effect in the heat-capacity prediction. This task was easily achieved by a simple algorithm which scanned each molecule for the first single bond having an oxygen atom at one end and a hydrogen atom at the other and, if found, skipped it. The exclusion of these molecules from the list of liquids yielded a distinctly better compliance with the experimental data as is demonstrated in the correlation diagram (Figure 4). The corresponding histogram (Figure 5) also shows a more symmetrical deviation distribution about the zero point. The good accordance of the experimental with the calculated Cp(liq.298) values, with a correlation coefficient R^2^ of 0.9890 and a MAPD of 6.51% based on a large variety of chemical structures within the 1102 compounds, allows great confidence in Equation (2), which translates the “true” volume Vm of a molecule free of hydroxy groups into its liquid heat capacity at 298.15 K.
Cp(liq.298)_OH-free_ = −5.3055 + 1.8183 × Vm(2)

Apart from the many ordinary compounds, for which the Cp(liq) at 298.15 K have been predicted in great accordance with experiment, the ILs have gained particular interest as a class of polar solvents, for which the knowledge of the liquid heat capacity is especially important. Hence, a list of the experimental and calculated data of the 145 ILs, which were included in the calculations of the parameters of Equation (2), are presented in Table 3. The mean values of the deviations shown at the bottom of the list indicate that Equation (2) tends to slightly underestimate liquid heat capacities by ca. 1.6%; however, in view of the relatively large “local” standard deviation of 24.49 J/mol/K or 5.14% (evaluated by means of the experimental and Equation (2)-predicted Cp values of Table 3), this deficiency seems acceptable. Beyond this, an independent direct correlation calculation with the molecular volumes and the experimental Cp(liq,298) of the ILs of Table 3 revealed a correlation coefficient of 0.9873 and a standard deviation R^2^ of 22.64 J/mol/K, values that are very similar to the ones received by means of Equation (2), confirming its applicability. In addition, in order to assess the cause of the fairly large “local” standard deviation of 24.49 J/mol/K for this class of compounds, an analogous correlation calculation with the homologous series of 1-CH_3_(CH_2_)_n_-3-methylimidazolium bis(trifluoromethanesulfonyl) amide, with n = 1–7, 9 and 11, was carried out, which should yield a steady increase of their liquid heat capacities by ca. 32.4 units per methylene group. The resulting standard deviation of 14.0 J/mol/K clearly points to experimental inaccuracy as the main cause of the large standard error.

A further drawback concerning this class of compounds is the relatively large number of outliers, defined as molecules, for which the difference between the experimental and predicted Cp value exceeds three times the standard deviation: more than half of the total number of 80 Cp(liq,298) outliers are ILs. In some cases, their experimental value is closer to—or even below—the predicted value for the heat capacity of their solid phase. The list of Cp(liq) outliers has been added to the Appendix A.

Siloxanes are a class of compounds that have become increasingly important for their thermodynamic stability at high temperatures as working fluids in the field of organic rankine cycle processes [204]. Table 4 compares the experimental liquid heat capacities at 298.15 K of 23 siloxanes with the values predicted by Equation (2). The mean deviation between the experimental and the calculated values, shown at the bottom of Table 4, exhibits a general underestimation by Equation (2) by 1.5%. An independent correlation calculation with the molecular volumes of the siloxanes of Table 4 and their experimental Cp(liq,298) values yielded a correlation coefficient R^2^ of 0.9866 and a standard deviation of 29.09 J/mol/K. While R^2^ is very similar to that resulting from Equation (2) (see Figure 2), the corresponding deviation is only moderately better than that listed at the bottom of Table 4. This, on the one hand, again confirms the general applicability of the present method on the calculation of the liquid heat capacities. The relatively large standard deviation, on the other hand, requires an answer as to its origin. Therefore, an analogous correlation calculation using the (incomplete) homologous series tetramethyoxy-, tetraethoxy-, tetrapropoxy-, tetrabutoxy-, tetraheptoxy-, tetraoctoxy- and tetradecoxysilane alone, which should show a correspondingly steady increase of their liquid heat capacities by ca. 123.8 units per four methylene groups, was carried out. This calculation indeed revealed a high correlation coefficient R^2^ of 0.9948, but still a large scatter with a standard deviation of 27.96 J/mol/K, allowing for the assumption that the large standard deviation listed in Table 4 is essentially caused by questionable experimental data.

Finally, in Table 5, the experimental liquid heat capacities of 222 hydrocarbons are compared with the predicted ones, evaluated by means of Equation (2). The mean values of the deviations added at the bottom show that the “global” Equation (2) on average overestimates the Cp(liq,298) values of the hydrocarbons by 6.34%. The predicted liquid heat capacities of the (incomplete) homologous series of 21 linear alkanes from butane to unatriacontane, on the other hand, are in excellent agreement with the experimental values, with the exception of eicosane and unatriacontane. Therefore, the reason for the general overestimation of the hydrocarbons and their standard deviation of 22.1 J/mol/K had to be found elsewhere.

A careful scan through Table 5 exhibits a distinct discrepancy of the deviations between the predictions and experimental Cp(liq,298) values of the noncyclic and the cyclic hydrocarbons: the deviations of the latter are nearly always much more negative, i.e., Equation (2) systematically and distinctly overestimates their liquid heat capacity. This is particularly clear on comparison of the linear alkanes as excellent benchmarks with the cycloalkanes having the same number of carbon atoms: the predicted values of the corresponding cyclic alkanes are systematically too high. A few examples may illustrate the difference in the deviation of the predictions: butane: 2.84% vs. cyclobutane: −13.26%; pentane: −0.01% vs. cyclopentane: −15.3%; hexane: 0.08% vs. cyclohexane: −12.14%; heptane: −1.1% vs. cycloheptane −13.23%; octane: −0.99% vs. cyclooctane: −10.29%. The same observation has been made with branched cycloalkanes, e.g., methylcyclopentane vs. hexane or dimethylcyclohexane vs. octane. As these deviations are systematic and therefore cannot be ascribed to experimental inaccuracies, they indicate an important limitation of the present prediction method: the “true” molecular volume does not adequately reflect the decrease of the rotational degrees of freedom within the cyclic moiety of a molecule in relation to a ring-open one, although the volume of a cyclic structure is necessarily smaller than a non-cyclic one with the same number of carbon atoms due to the deduction of the partial volume of two hydrogen atoms per cycle which, however, only corresponds to a difference of 17–23 J/mol/K per cycle between cyclic and non-cyclic alkanes, depending on the molecule size. 

A weaker trend of this kind can be found when comparing linear with branched alkanes of the same chemical formula in that the branched species have systematically and significantly lower experimental heat capacities than their linear relatives, although their molecular volumes are all of nearly equal size, and thus Equation (2) would suggest very similar Cp values. Some examples of the differences of deviation may be given: pentane: −0.01% vs. 2,2-dimethylpropane (neopentane): −9.01%; heptane: −1.10% vs. 3,3-dimethylpentane: −5.40%. Hence, it seems that branching also lowers the number of effective rotational and possibly vibrational degrees of freedom. Nevertheless, these prediction errors rarely exceed the methods’ standard deviation but should be kept in mind when applying Equation (2).

#### 3.4.2. Heat Capacity of Liquid Alcohols and Acids

The exceptionally high polarity of the hydroxy group has been shown in the previous subsection to exhibit a decisive, enhancing effect on the heat capacity of a molecule. Figure 6 demonstrates the correlation of the molecular volume with the experimental liquid heat capacity of the 194 alcohols and acids extracted from the complete set used in Figure 2, revealing, apart from a fairly large scatter, some peculiarities in the volume range of 135 to 190 A^3^ that require an explanation. Beyond this, the obvious asymmetry of the corresponding histogram (Figure 7) hints at a similar deficiency of this simple prediction method as in the previous subsection, this time disclosing a systematic underestimation of the molecules carrying two or more OH groups. This inadequacy was resolved by separating the molecules carrying a single OH group from those carrying two or more. The results are shown in the correlation diagrams in Figure 8 and Figure 9. The corresponding lists of molecules with their molecular volumes, experimental and predicted data as well as their deviations are added as Table 6 and Table 7.

The corresponding Equation (3) for the heat capacity (Cp(liq.298)_OH1_) of the monoalcohols and –acids and Equation (4) for the heat capacity (Cp(liq.298)_OH>1_) of the polyalcohols and –acids are therefore as follows, wherein Vm is the “true” molecular volume:Cp(liq.298)_OH1_ = 23.3101+ 1.9282 × Vm(3)
Cp(liq.298)_OH>1_ = 23.5782 + 2.0422 × Vm(4)

As mentioned, at the beginning of this subsection, Figure 6 and Figure 8 reveal an abnormality in the volume range at ca. 135 and 190 A^3^, in that the experimental Cp(liq,298) values of clusters of compounds with similar molecular volume distinctly deviate from the predicted values, hinting at a systematic deficiency of the present prediction method as already discussed with the OH-free compounds. In order to assess the importance of these deviations, it is helpful to start from a common base. In this case, in analogy to the linear alkanes used in the prior subsection, the linear alcohols may serve as the starting point. Scanning Table 6, the experimental values of this group of alcohols systematically deviate from the predictions by, on average, −10.75 J/mol/K (the value for 1-hexadecanol has been omitted as being an obvious outlier with −54.06 J/mol/K), i.e., Equation (3) systematically overestimates their Cp(liq,298) values. A similar overestimation is found with the branched 1-alkanols in Table 6, e.g., 2-methyl-1-propanol, 2-ethyl-1-butanol, 3,3-dimethyl-1-butanol and 2-methyl-1-pentanol. An even larger overestimation of, on average, −18.58 J/mol/K is found for the four cyclic alcohols cyclobutanol, cyclopentanol, cyclohexanol and cycloheptanol. On the other hand, with the exception of 2-propanol the experimental Cp(liq,298) values of all the 16 unbranched secondary alcohols of Table 6, i.e., 2-butanol, 3-pentanol, 2- and 3-hexanol, 2-, 3- and 4-heptanol, 2-, 3- and 4-octanol, 2-, 3-, 4- and 5-nonanol and 5-decanol, are higher by an average of +13.16 J/mol/K than the predicted ones. Even worse is the deviation of the 17 branched secondary alcohols in Table 6 with an average of +14.51 J/mol/K. However, the largest underestimation by Equation (3) with an average deviation of +35.82 J/mol/K was found for the 8 tertiary alcohols 2-methyl-2-propanol, 2-methyl-2-butanol, 2-methyl-2-pentanol, 3-methyl-3-pentanol, 2-Methyl-2-hexanol, 2-methyl-2-heptanol, 4-methyl-4-heptanol 4-propyl-4-heptanol of Table 6.

Comparison of these findings with those discussed with the alkanes of the prior subsection yields one accordance and one striking contradiction: accordance is found with the cyclic compounds in that in both cases Equation (3) overestimates the experimental Cp(liq,298) values. On the other hand, the results of the unbranched and branched secondary and tertiary alcohols are counterintuitive in that they all may be viewed as analoga of the branched alkanes and, thus should also have lower liquid heat capacities than their linear relatives. Serra et al. [73] explained the discrepancies of the heat capacities of a series of C7 alcohols by the strength of the hydrogen bridges, which they assumed to be dependent on the steric hindrance of the hydroxy group. This steric hindrance is lowest with primary alcohols and highest with tertiary alcohols, as is demonstrated in Figure 10. Hence, the hydrogen bridging potential should increase in the order tertiary < secondary < cyclical < primary alcohols. The findings in Table 6 obviously oppose this order because it is common knowledge that the stronger the hydrogen bridges, the higher the heat capacity, e.g., compare water ((Cp(liq,298) = 75.32 J/mol/K) with ammonia ((Cp(liq,298) = 33 J/mol/K). The resolution for this conflict might have been provided by the theoretical studies of Huelsekopf and Ludwig [288], who demonstrated, by means of the quantum cluster equilibrium theory (QCE), exemplified on two primary (ethanol and benzyl alcohol) and a tertiary alcohol (2,2-dimethyl-3-ethyl-3-pentanol), that the primary alcohols principally exist as cyclic tetramers and pentamers in the liquid phase, whereas the tertiary alcohol only forms monomers and dimers. In other words, the higher liquid heat capacity of the secondary and tertiary alcohols could be owed to the formation of smaller clusters, which overall leads to a higher number of rotational and translational degrees of freedom. This could explain the systematic overestimation of the liquid heat capacity of the 1-alkanols by Equation (3) in that in these cases the experiments were probably carried out on cyclical tetra- or pentamers exhibiting lower translational and rotational freedoms than smaller clusters or even monomers. It could also explain the exceptionally low heat capacity of the saturated cyclic alcohols: as demonstrated in Figure 10, the hydroxy group in cyclic alcohols is sterically hardly hindered and, thus could also form cyclic tri-, tetra- or pentamers, reducing the number of motional degrees of freedom in addition to the reduction of the degrees of freedom due to cyclic skeleton of the compound itself as discussed for the cycloalkanes. Carboxylic acids are known to build dimers in the liquid phase. Hence, one would expect—following the arguments for the linear alcohols—that their liquid heat capacity should again be generally lower than calculated. This is indeed the case: the mean deviation from the calculated values is −19.57 J/mol/K for the 11 linear monocarboxylic acids from acetic acid to dodecanoic (lauric) acid listed in Table 6. All these systematic deviations between experiment and prediction in the classes of alcohols and acids demonstrate a fundamental limit of the present prediction method in that it can principally not consider intermolecular effects on the liquid heat capacity. This reflects the findings of Ruzicka and Domalski [7], who stated that the “group of oxygen compounds includes families, such as alcohols and aldehydes, that exhibit the largest prediction error of all families of organic compounds” by their second-order GA method. However, they did not elaborate on the reason.

For comparison, the correlation statistics of the OH-free compounds (Figure 4) with those of the alcohols and acids before separation (Figure 6), and those of the mono-alcohols and –acids (Figure 8) as well as those of the poly-alcohols and –acids (Figure 9) have been collected in Table 8. The difference in the heat capacities of the OH-free molecules and the alcohols and acids appears very prominently in the values of the intercepts and the slopes. Comparison of these two values clearly shows the large enhancing effect of the OH groups on the liquid heat capacity of the alcohols and acids. In addition, Table 8 also reveals a close similarity between the set of alcohols and acids before separation and that of the monoalcohols and –acids. Finally, the largest intercept and slope for the polyalcohols and –acids indicate that this group generally exhibits the highest Cp(liq,298) values. The exceptionally high correlation coefficient of 0.991 for this group, however, should not be overrated, as it mostly consists of linear, α,o-substituted, primary alcohols and acids. (The sum of the mono- and polyalcohols/acids only adds up to 200; the missing 201st compound of Figure 6 is water, which has not been included in the separated sets). 

### 3.5. Heat Capacity of Solids

While true liquid phases of molecules are isotropic and thus appear as only one single phase in the measurement of their heat capacity, solids require special care with respect to the association phase in which the molecules are arranged at room temperature. In many cases molecules crystallize in several, energetically different phases, which not only have different heat capacities but can also change from one phase into another one at the time of the heat capacity measurement. Beyond this, quite often the molecules, although in a seemingly solid form, have not really crystallized but are in truth a supercooled melt. These uncertainties may be a major reason for the larger scatter of the Cp(sol,298) compared to the Cp(liq,298) values measured by independent sources as referenced by Chickos et al. [5]. Irrespective of these difficulties, the heat capacity of solids should always exhibit a lower value than that of liquids due to the inhibition of the translational and rotational freedoms of motion of the molecules in the crystalline phase. In order to compare the heat capacities of solids with those of liquids, the total of 797 compounds with known experimental Cp(sol,298) data have been separated into an OH-free set and one that encompasses all alcohols, sugars and acids. The correlation of the former set is shown in Figure 11, that of the latter in Figure 12. The OH-containing 241 compounds have been further separated into 123 monools (Table 9) and monoacids and into 118 polyols and polyacids (Table 10). Their corresponding correlations are visualized in Figure 13 and Figure 14. 

The parameters for the final Equations (5)–(7) for the prediction of the solid heat capacities correspond to the intercepts and the slopes of the regression lines resulting from the correlations in Figure 11, Figure 13 and Figure 14. In these equations the variable Vm and the subscripts at the heat capacities Cp(sol,298) have the same meanings as given for Equations (2)–(4):Cp(sol.298)_OH-free_ = 2.8899 + 1.3669 × Vm(5)
Cp(sol.298)_OH1_ = −11.2179 + 1.5050 × Vm(6)
Cp(sol.298)_OH>1_ = −14.9656 + 1.5462 × Vm(7)

Figure 11 demonstrates the good correlation of the “true” molecular volume with the experimental solid heat capacity of compounds of all OH-free classes, i.e., excluding alcohols, sugars and acids. Among these classes, the metallocenes [196,197,198,199] in Table 11 are especially interesting in that they demonstrate the inertness of the heat capacity towards the central atom and its McGowan-vdw radius [42], which varies from 0.74 A for Fe^2+^ to 0.99 A for Mn^2+^, because the metal ions are nearly completely encapsulated by the two cyclopentadienyl ligands. 

In Table 12, the solid heat capacities of the siloxanes [201,202,203,204] are collected for comparison with the values of their liquid heat capacities in Table 4. The experimental Cp(sol,298) values are on average underestimated by 10.64 J/mol/K by Equation (5). An independent Vm vs. Cp(sol,298) correlation calculation of these 16 siloxanes yielded a correlation coefficient of 0.9778 and a standard deviation of 38.29 J/mol/K, confirming the fairly large scatter.

Table 13 presents a list of the hydrocarbons for which experimental solid heat capacities were available. Their mean deviation, shown at the bottom of Table 13, again indicates a general underestimation of the Cp(sol,298) values for the hydrocarbons by Equation (5) by ca. 4%. (An independent calculation, based on the hydrocarbons of Table 13 only, resulted in a correlation coefficient of 0.9779, a standard deviation of 28.35 J/mol/K and a MAPD of 6.96%. The intercept of the regression line was calculated to −21.9433 and the slope to 1.4291). The standard deviation of their experimental values from the predicted ones is much larger than that of their liquid heat capacity shown in Table 5, which could be ascribed to uncertainties such as various crystal forms mentioned at the beginning of this subsection. Some examples may shed some light on their impact: the three structural isomers o-, m- and p-quinquephenyl have nearly the same molecular volume of 364.1 A^3^, resulting in a predicted Cp(sol,298) value of 500.6 J/mol/K. Yet, the experimental values are given as 444.3, 443.7 and 455.5 J/mol/K, respectively, i.e., their values deviate by up to 11.9 units. Similarly, for the three isomers o-, m- and p-terphenyl, a solid heat capacity was calculated to 308.7 ± 0.7 J/mol/K; the experimental values are 274.75, 281.0 and 279.6 J/mol/K, respectively, i.e., a difference of up to 6.25 units. Evidently, in both cases the ortho-isomer has a helical structure, in contrast to the more planar structure of the m- and p-isomers, which probably leads to a crystal structure that differs from that of the m- and p-isomer. For anthracene and phenanthrene, two very closely related compounds, the present prediction method suggests Cp(sol,298) values of 234.9 and 233.8 J/mol/K; the experimental values are 210.5 and 220.3, respectively, i.e., a difference of 9.8 units. Finally, for the three isomers 2,3-, 2,6- and 2,7-dimethylnaphthalene Cp(sol,298) values of 219.8, and twice 220.5 J/mol/K, respectively, have been predicted; the experimental values are 216.47, 204.39 and 204.20 J/mol/K, a difference of up to 12.27 units. Nevertheless, these deviations never exceeded the standard deviation of 23.66 J/mol/K.

In Table 14, the correlation statistics data of the heat capacities of solids have been collected. In this table, the differences between the heat capacities of the solid OH-free compounds and those of the solid alcohols, acids and sugars are again centered on the parameters of the regression lines, i.e., the intercepts and slopes, analogous to the ones discussed in the case of the liquids (Table 8). Comparing the corresponding parameters in Table 8 for liquids with those in Table 14 for solids and applying them in the respective Equations (2)–(7), it is immediately evident that the heat capacities of solids calculated in this way are indeed always smaller than those of liquids. A rough calculation, however, comparing the results of Equations (5)–(7), e.g., using an average molecular volume, reveals much smaller differences of the solid heat capacities between the OH-free compounds and the alcohols or acids than when applying Equations (2)–(4) in the case of their liquid heat capacities. This is in accord with the notion that hydrogen bridges in crystals have no significantly additional effect on the inherently restricted freedoms of motion. Their contribution is essentially of the vibrational type.

Equations (2)–(7) enable an estimate of the heat-capacity differences between the liquid and solid phase for the three classes of compounds by simply subtracting the respective equations for the liquid phase from the ones for the solid phase. Thus, for the OH-free class of compounds, the rounded difference ΔCp(298)_Oh-free_ is calculated by Equation (8), and for the other classes the corresponding differences are defined by the Equations (9) and (10):ΔCp(298)_OH-free_ = −8.20 + 0.45 × Vm(8)
ΔCp(298)_OH1_ = 34.53 + 0.42 × Vm(9)
ΔCp(298)_OH>1_ = 38.54 + 0.50 × Vm(10)

The comparison of Equation (8) with Equation (9) immediately shows the large median additional effect of a hydroxy group on the heat capacity upon the phase change of a molecule from the solid into the liquid phase, exemplified by two compounds of similar molecular volume: anisole (109.1 A^3^)) and o-cresol (108.5A^3^). For both compounds a solid heat capacity of ca. 152 J/mol/K was predicted (for o-cresol the experimental value is 154.56 J/mol/K [48]). For anisole the calculated liquid Cp value was 193.1, for o-cresol 232.7 J/mol/K. These values have been confirmed by the experimental data: for anisole a value of 199 J/mol/K [47] was published, for o-cresol 229.75 J/mol/K [48]. Equation (10) indicates that further hydroxy groups provide a substantially lower contribution.

At the beginning of this subsection, the subject of the definition of the solid phase of a molecule under experimental conditions has been mentioned, which adds an uncertainty to the experimental value of the solid heat capacities. Beyond this, since the present calculation method allows the prediction of both the solid and liquid heat capacity of any molecule, it can be demonstrated that in several cases the experimental value of the alleged solid heat capacity of a molecule is much more closely related to that of its liquid phase. This may be illustrated by two examples: for benzylideneaniline, a Cp value of 302.67 J/mol/K was measured [163]; the solid heat capacity was predicted at 244.5 J/mol/K and the liquid one at 316.1 J/mol/K. In view of its melting point of 54 °C which is fairly close to the experimental conditions, the assumption is not unrealistic that the compound could at least be partially molten. For 2-methyl-3-amino-4-methoxymethyl-5-aminomethylpyridine, the experimental—supposedly solid—heat capacity was published as 307.1 J/mol/K [48]; the corresponding solid and liquid Cp values have been calculated to 240.3 and 310.5 J/mol/K, respectively. A melting point has not been given. A number of outliers, which had to be excluded from the solid heat-capacity calculations and are separately listed in the Appendix A, would have suited very well in the liquid Cp calculations and vice versa. In fact, for several molecules the experimental solid Cp value was even higher than their calculated liquid Cp value. To set the experimental values in relation to the predicted ones, both the calculated solid and liquid Cp data have been added to the liquid and solid outliers lists.

### 3.6. Temperature Dependence of the Heat Capacities

The great majority of the publications cited in Section 3.3, particularly the more recent ones, not only present heat capacity data at specific temperatures but also provide temperature profiles over a certain range, e.g., from below the molecule’s melting up to its boiling point. Considering the evident linearity of the correlations between the “true” molecular volumes and the Cp values at the ultimately arbitrary 298.15 K, it was obvious to try to find analogous correlations at different temperatures. In order to receive a representative picture of the influence of the temperature on the correlations, two temperatures below and two above the standard value have been chosen at a space of 25 K, resulting in a set of the four temperatures at 250, 275, 325 and 350 K. Cp values not directly listed at these temperatures in the publications have been linearly interpolated by means of values listed at the nearest two temperatures. This set was then applied on the group of OH-free and OH-carrying compounds in their liquid and solid phase. (It quickly turned out that a separation of the OH-carrying group of compounds into a mono- and a polyhydroxy subgroup was either not feasible due to the insufficient number of examples or had a negligible impact on the results). Figure 15, Figure 16, Figure 17 and Figure 18 demonstrate the correlation diagrams at the mentioned temperatures for the liquid and solid phase of the OH-free and OH-carrying sets of compounds. In Table 15, Table 16, Table 17 and Table 18 their respective statistical data have been collected and combined with the statistics data of the correlation at the standard temperature.

Figure 15, Figure 16, Figure 17 and Figure 18 prove that linearity of the Vm vs. Cp correlations is independent of temperature. Figure 16B–D, showing the liquid heat capacities of OH-carrying compounds at 275, 325 and 350 K, exhibit the largest scatter of data and thus yielded the poorest correlation coefficients (see Table 16), reflecting the observation of the strong effect on the intermolecular hydrogen bonds on going from primary to secondary and tertiary alcohols discussed earlier. Looking at Table 15, Table 16, Table 17 and Table 18, a common feature is immediately apparent: the slope of the regression lines always increases with the increasing temperature. This seems logical considering that compounds of any molecular volume have no heat capacity at all at 0 K, and thus the slope would have a value of zero, and that, on the other hand, with both the growing molecular volume as well as the rising temperature the number of degrees of freedom of motion and vibration increases, thus multiplying their increasing effect on the heat capacities. Analogous to the Equations (2)–(7), Table 15, Table 16, Table 17 and Table 18 enable the calculation of the solid and liquid heat capacities at 250, 275, 325 and 350 K, using the corresponding general Equation (11), wherein T is the selected temperature in K, the values of the intercept and slope are given for this temperature in Table 15, Table 16, Table 17 and Table 18, and Vm is the “true” molecular volume:Cp(T) = intercept + slope × Vm(11)

In Table 19 and Table 20 a number of examples demonstrates the expandability of the present Cp-prediction method to a range of temperatures. For many compounds, nearly perfect linearity of the temperature dependence of the heat capacity in the range between 250 and 350 K has been graphically demonstrated (specifically for hexatriacontane [51], alkylsubstituted adamantanes [53], neopentylbenzene [54], 4,4′-disubstituted biphenyls [59], tetracene and pentacene [62], 2-propenol and cyclohexylalcohols [76], adamantanols [77], monoterpenoids [85], a,ω-alkanediols [88], 1,2-cyclohexanediol [99], ribose and mannose [100], ketohexoses [101], glucose [102], sugar alcohols [103], 3,5-di-t-butylsalicylic acid [111], 2-pyrazinecarboxylic acid [116], vitamin B3 [118], 2,4-dinitrobenzaldehyde [145], various monoterpenes [147,148], 2-pyridinealdoxime [164], chloroanilines and chloronitrobenzenes [166], linear alkyldiamides [170], 2-thiobarbituric acids [175], monuron [179], 1,3,5-trithiane [189], ferrocene derivatives [198,199], cyclic siloxanes [202], adenosine [206], tryptophan [210], carnitine [211], 2-(chloromethylthio)benzothiazole [231], 2-amino-5-nitropyridine [233], 2-aminopyridine [234], 4-dimethylaminopyridine [236], 8-hydroxyquinoline [237], caffeine [238], 4′-bromomethyl-2-cyanobiphenyl [249], myclobutanil [250], fenoxycarb [251], methylprednisolone [254], *N*-methylnorephedrine [255], *N*,*N*-dimethylnorephedrine hydrochloride [256], risperidone [258], and vitamin B2 [259]). Table 19 and Table 20 have made use of this linearity in that the predicted values of some intermediate temperatures have been linearly interpolated using the two values of the nearest temperatures, calculated by means of Equation (11) and the corresponding parameters of Table 15, Table 16, Table 17 and Table 18.

Table 19, presenting 1-phenylpyrazole as an example of the OH-free compounds and 1-octanol representing the OH-carrying molecules class for the calculation of their liquid heat capacities, very clearly shows the difference in the reliability of the correlations of these two classes: the Cp predictions for the alcohol reveal larger deviations over all temperatures than for the OH-free compound, reflecting the larger scatter in the Figure 16D and the correspondingly lower correlation coefficients in Table 16. The representative of the OH-free class in Table 20, 3,4-dimethoxyphenylacetonitrile, has a melting point of 311.82 K (38.67 °C). Hence, the calculated solid heat capacity at 325 K is fictional, because at this temperature the compound has probably mostly changed to the liquid phase. However, its calculation enabled the interpolation of the Cp(sol) data at 310 and 320 K, which reveals by the increasing deviation from the experimental values that these are already increasingly “contaminated” by the energy absorption caused by the phase change process.

## 4. Conclusions

This paper has presented a universally applicable method for the calculation of the heat capacities of the solid and liquid phase of all molecules, based on a property that is common to every molecule, its molecular volume. The main advantage of the present approach lies in its absence of any requirement of further experimental data to evaluate a molecule’s “true” volume, which allows the extension of the correlation results between molecular volumes and known experimental heat capacities for liquids and solids for the heat-capacity prediction of any further imaginable molecule. Therefore, in project ChemBrain IXL, the predicted Cp data for liquids and solids have been routinely added to all of the presently ca. 31’600 compounds. The enablement to predict both the heat capacities of the liquid as well as the solid phase even for molecules for which at standard conditions only one of them is experimentally accessible, e.g., in borderline cases, allows an assessment as to whether an examined compound is really present in a defined crystalline form or e.g., rather a super-cooled melt. The possibility to cover the entire scope of compounds by means of just two parameters of a linear regression line has also enabled the discovery of structural effects that have a strong influence on the accuracy of the prediction, two of which have been outlined in detail: (1) Cyclisation and branching, demonstrated with alkanes, diminishes the heat capacity of liquids. (2) The strong influence on the liquid heat capacity of the intermolecular hydrogen bonds with alcohols and acids as well as the restrictive effects of steric hindrance on the hydroxy group has been demonstrated. Despite the observation that these structural influences and the direction and magnitude of their effect are generally within the range of one or two standard deviations, they have to be taken into account on assessing the predicted values. Furthermore, it has been shown that the linearity of the correlation between the molecular volume and the heat capacity is not limited to the standard temperature, which also enables a reasonably reliable prediction of the temperature dependence of the heat capacities, at least in the vicinity of the standard temperature. The presented prediction approach evidently does not allow for a simple paper-and-pencil Cp calculation as described for other thermodynamic and further properties [30,31,32,33]. However, the computer algorithm for the calculation of the “true” molecular volume and the subsequent evaluation of the heat capacities by means of one of the Equations (2)–(7) is very simple and thus easily integrable in software dealing with 3D-molecular structures. 

The present work is part of an ongoing project called ChemBrain IXL available from Neuronix Software (www.neuronix.ch, Rudolf Naef, Lupsingen, Switzerland).

## Figures and Tables

**Figure 1 molecules-24-01626-f001:**
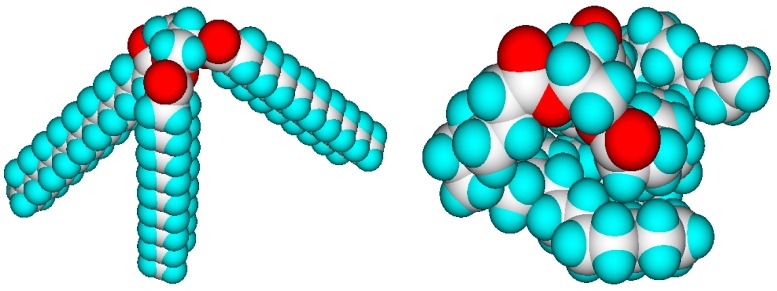
Energy-minimized forms of tristearin (graphics by ChemBrain IXL). (**Left**) stearyl chains stretched, Vm = 997.6 A^3^; (**right**) stearyl chains randomly folded, Vm = 992.9 A^3^.

**Figure 2 molecules-24-01626-f002:**
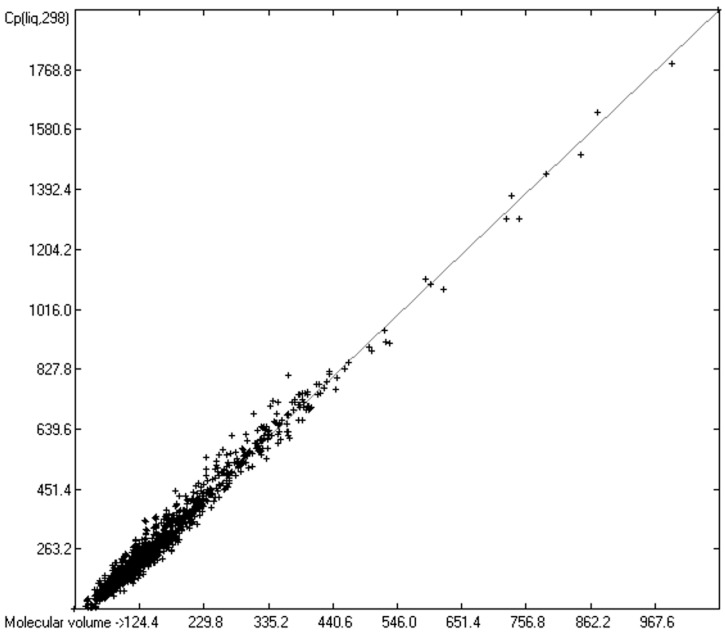
Correlation of molecular volumes with Cp(liq) at 298.15 K. (N = 1303, R2 = 0.9785, σ = 27.84 J/mol/K, MAPD = 8.23%, regression line: intercept = 1.7887, slope = 1.8211).

**Figure 3 molecules-24-01626-f003:**
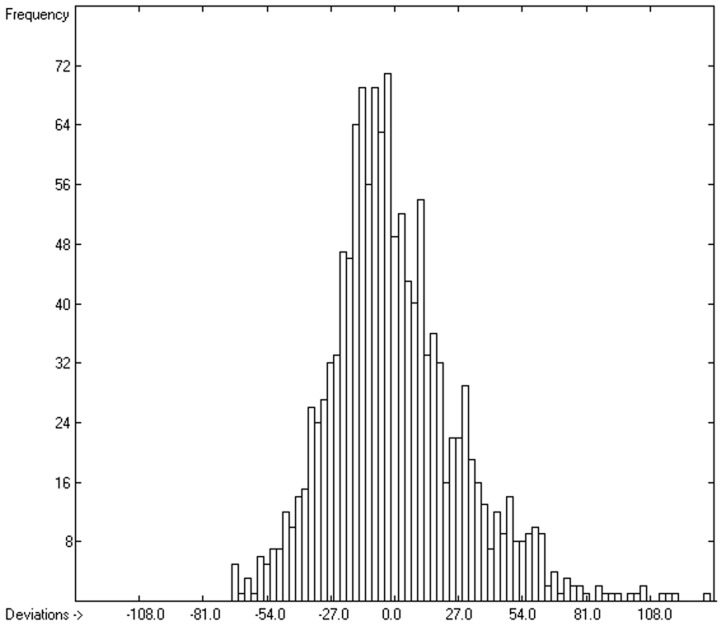
Histogram of molecular volumes with Cp(liq) at 298.15 K. Values range from 75.32 to 1956.10 J/mol/K.

**Figure 4 molecules-24-01626-f004:**
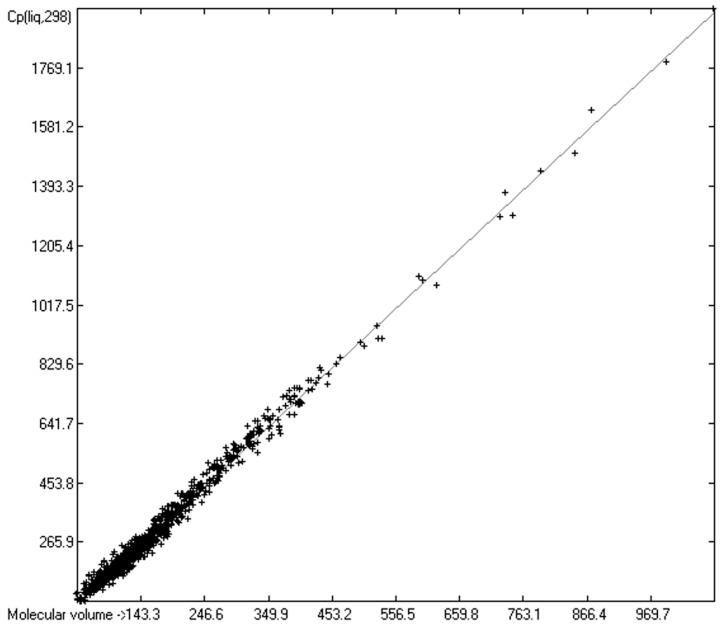
Correlation of molecular volumes with Cp(liq) at 298.15 K, excluding alcohols and acids. (N = 1102, R^2^ = 0.9890, σ = 20.70 J/mol/K, MAPD = 6.51%, regression line: intercept = −5.3055, slope = 1.8183).

**Figure 5 molecules-24-01626-f005:**
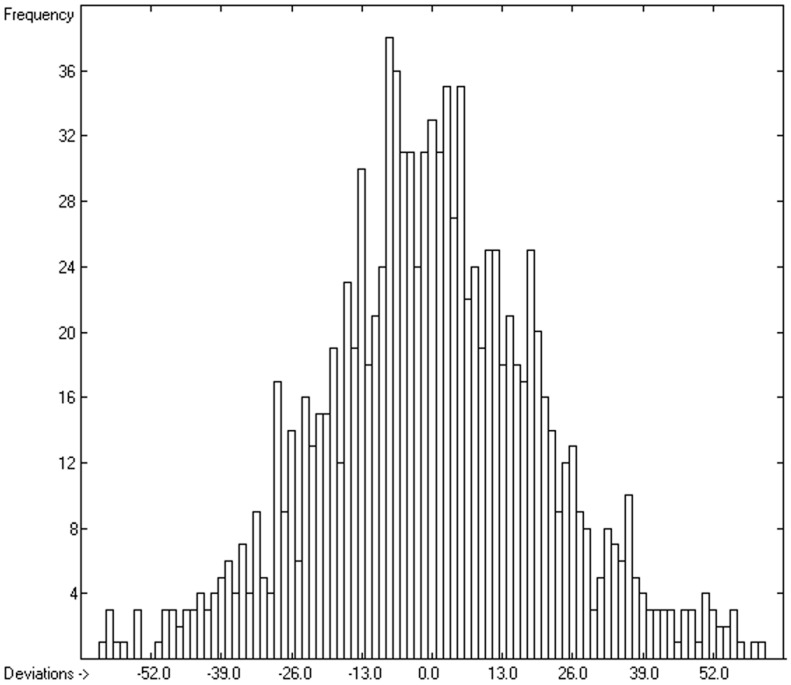
Histogram of molecular volumes with Cp(liq) at 298.15 K, excluding alcohols and acids. Values range from 78.7 to 1956.10 J/mol/K.

**Figure 6 molecules-24-01626-f006:**
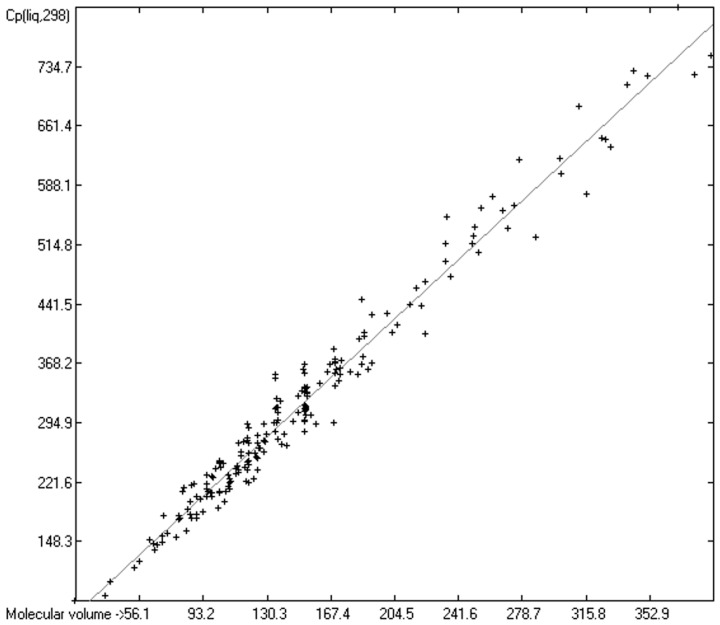
Correlation of molecular volumes with Cp(liq) at 298.15 K of mono- and polyols and –acids. (N = 201, R^2^ = 0.9724, σ = 23.24 J/mol/K, MAPD = 6.11%, regression line: intercept = 20.9141, slope = 1.9676).

**Figure 7 molecules-24-01626-f007:**
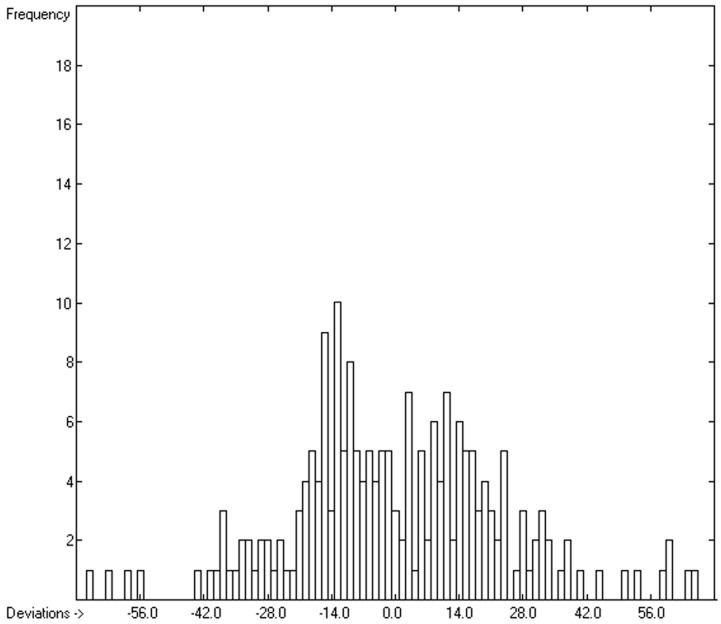
Histogram of molecular volumes with Cp(liq) at 298.15 K of 202 mono- and polyols and –acids. Value range from 75.32 to 807.5 J/mol/K.

**Figure 8 molecules-24-01626-f008:**
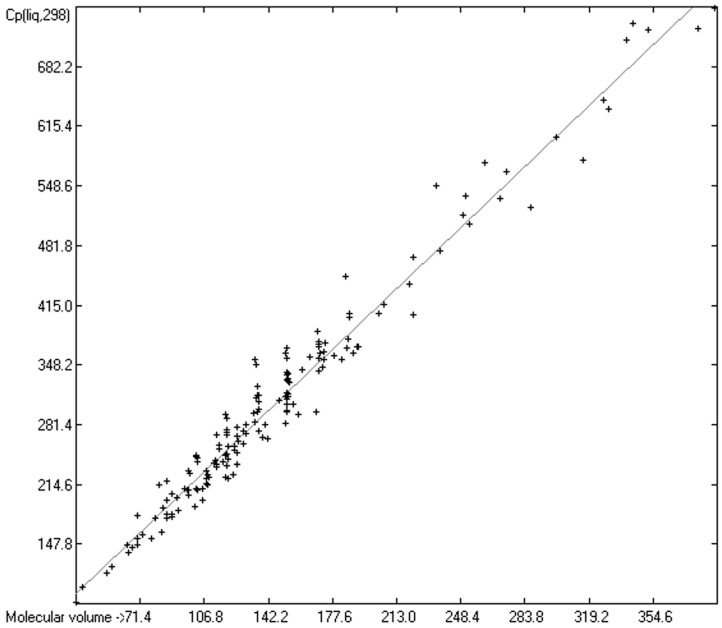
Correlation of molecular volumes with Cp(liq) at 298.15 K of monools and monoacids. (N = 164, R^2^ = 0.9685, σ = 22.91 J/mol/K, MAPD = 6.04%, regression line: intercept = 23.3101, slope = 1.9282, Value range from 81.92 to 748.0 J/mol/K).

**Figure 9 molecules-24-01626-f009:**
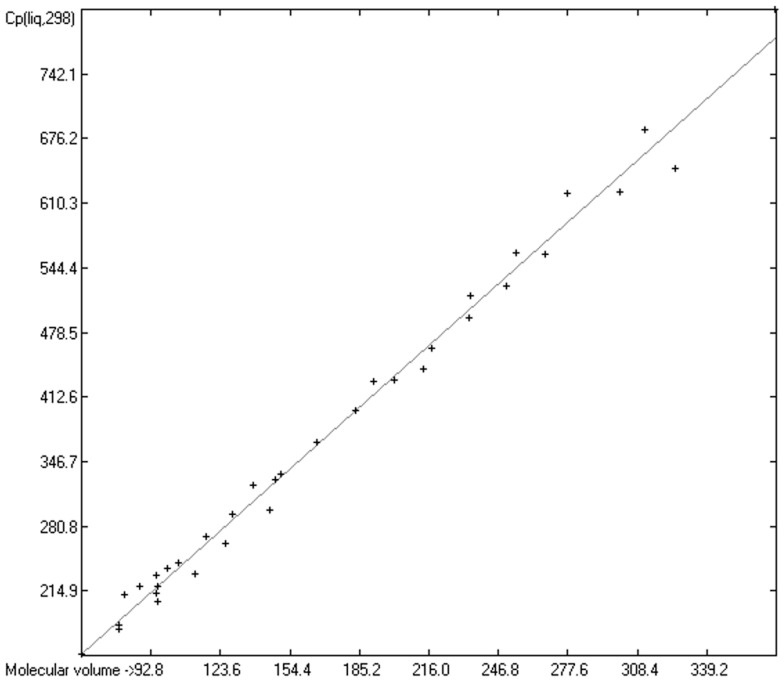
Correlation of molecular volumes with Cp(liq) at 298.15 K of polyols and polyacids. (N = 36, R^2^ = 0.9910, σ = 16.03 J/mol/K, MAPD = 3.77%, regression line: intercept = 23.5782, slope = 2.0422, Value range from 149.8 to 807.5 J/mol/K).

**Figure 10 molecules-24-01626-f010:**
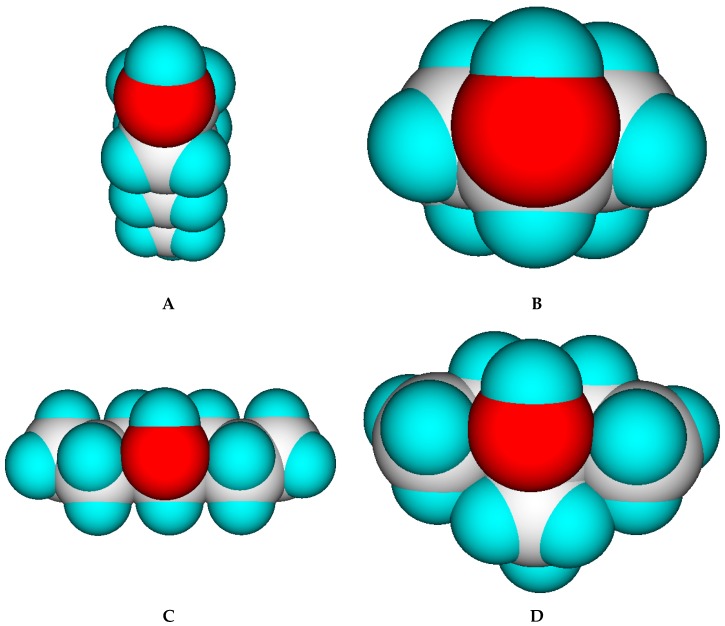
Top view of energy-minimized forms of four C7 alcohols (graphics by ChemBrain IXL). (**A**): 1-heptanol. (**B**): 4-methylcyclohexanol, (**C**): 4-heptanol. (**D**): 3-ethyl-3-pentanol.

**Figure 11 molecules-24-01626-f011:**
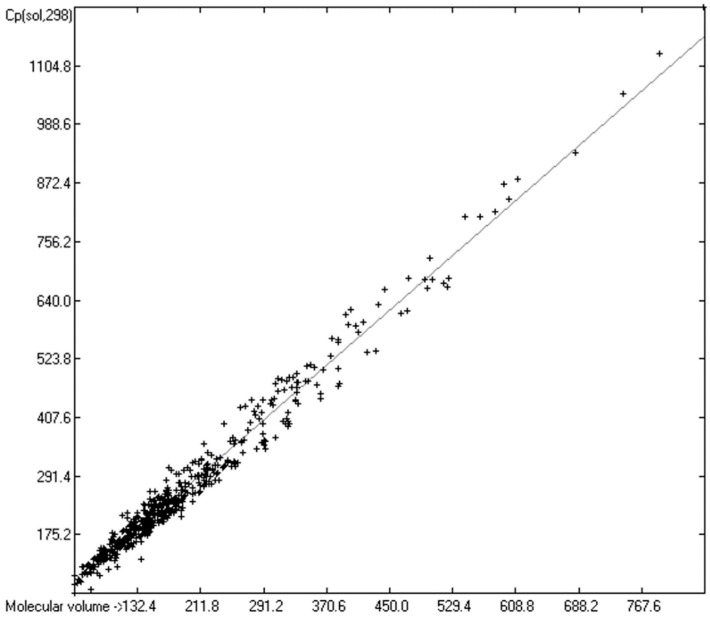
Correlation of molecular volumes with Cp(sol) at 298.15 K of OH-free compounds. (N = 555, R^2^ = 0.9766, σ = 23.14 J/mol/K, MAPD = 7.19%, regression line: intercept = 2.8899, slope = 1.3669, value range from 59.4 to 1220.9 J/mol/K).

**Figure 12 molecules-24-01626-f012:**
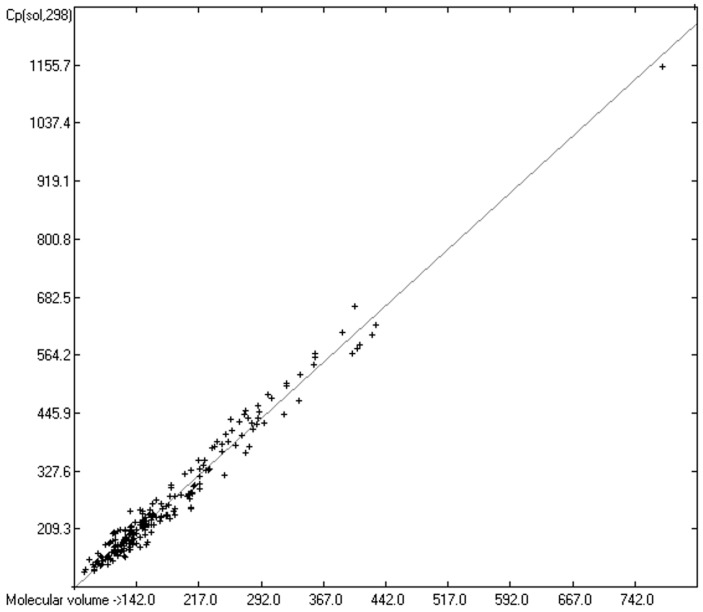
Correlation of molecular volumes with Cp(sol) at 298.15 K of mono- and polyols and -acids. (N = 242, R^2^ = 0.9792, σ = 20.87 J/mol/K, MAPD = 6.72%, regression line: intercept = −14.7861, slope = 1.5364, value range from 91 to 1273 J/mol/K).

**Figure 13 molecules-24-01626-f013:**
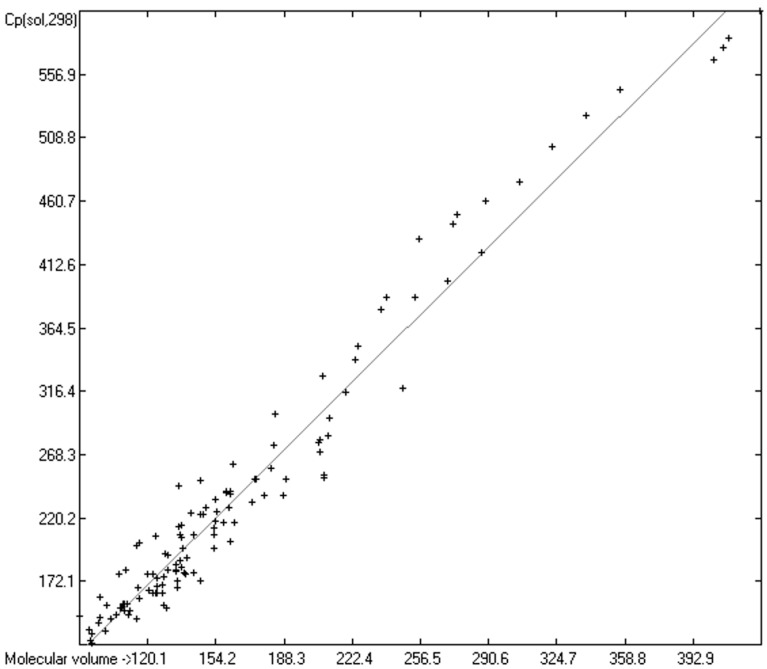
Correlation of molecular volumes with Cp(sol) at 298.15 K of monools and monoacids. (N = 123, R^2^ = 0.9613, σ = 21.62 J/mol/K, MAPD = 7.34%, regression line: intercept = −11.2179, slope = 1.5050, value range from 124.7 to 604.8 J/mol/K).

**Figure 14 molecules-24-01626-f014:**
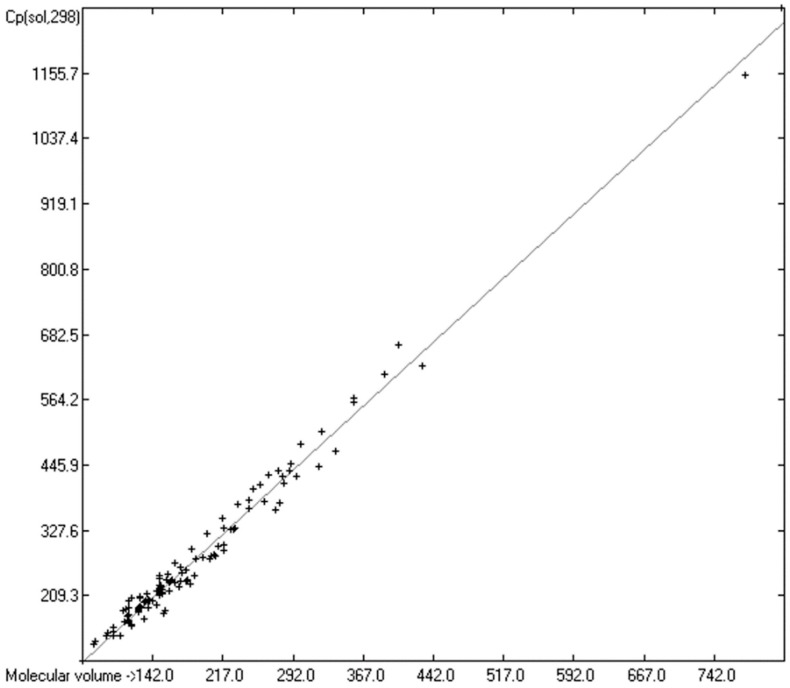
Correlation of molecular volumes with Cp(sol) at 298.15 K of polyols and polyacids. (N = 119, R^2^ = 0.9866, σ = 19.75 J/mol/K, MAPD = 5.93%, regression line: intercept = −14.9656, slope = 1.5462, value range from 91 to 1273 J/mol/K).

**Figure 15 molecules-24-01626-f015:**
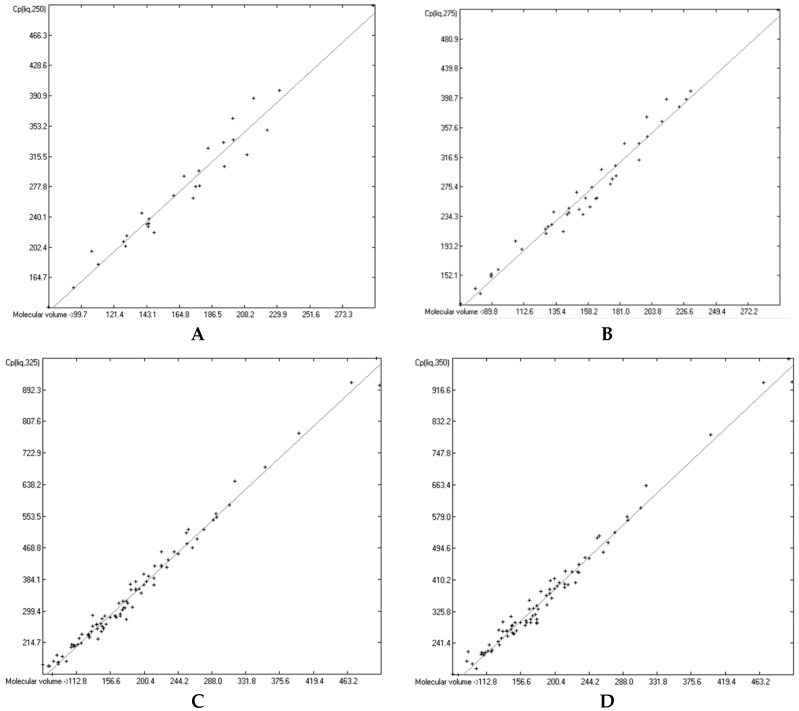
Correlation of molecular volumes with liquid heat capacities of OH-free compounds at various temperatures. (**A**): 250 K. (**B**): 275 K. (**C**): 325 K. (**D**): 350 K.

**Figure 16 molecules-24-01626-f016:**
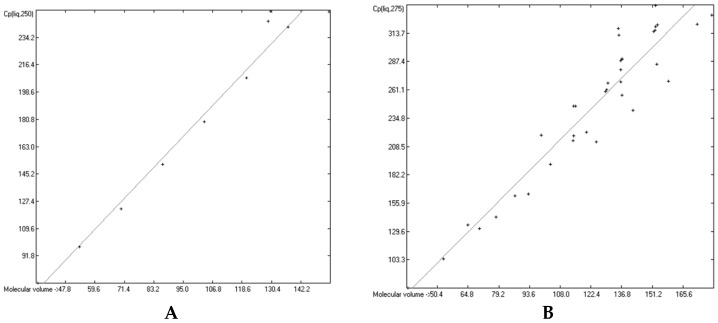
Correlation of molecular volumes with liquid heat capacities of OH-carrying compounds at various temperatures. (**A**): 250 K. (**B**): 275 K. (**C**): 325 K. (**D**): 350 K.

**Figure 17 molecules-24-01626-f017:**
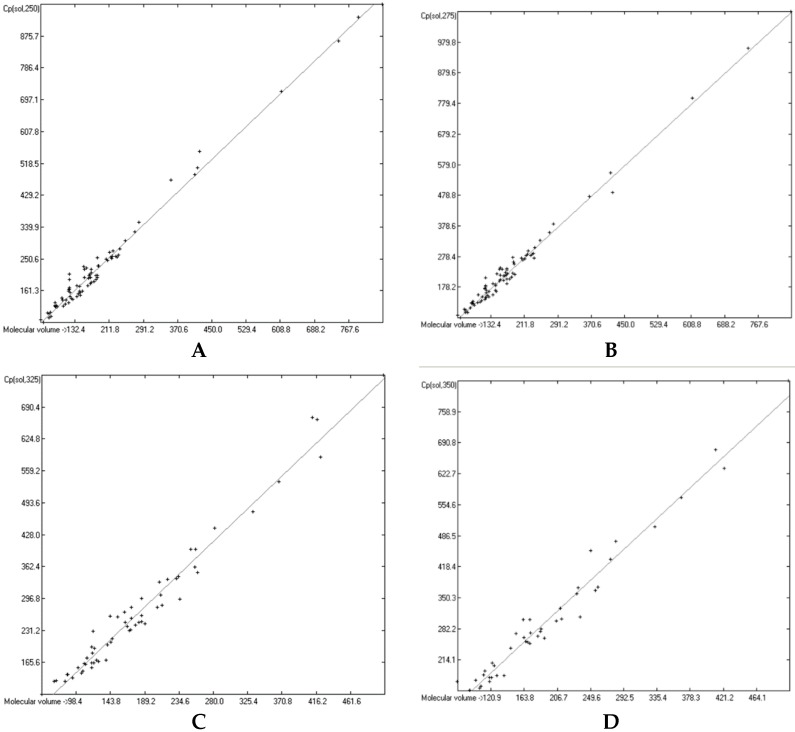
Correlation of molecular volumes with solid heat capacities of OH-free compounds at various temperatures. (**A**): 250 K. (**B**): 275 K. (**C**): 325 K. (**D**): 350 K.

**Figure 18 molecules-24-01626-f018:**
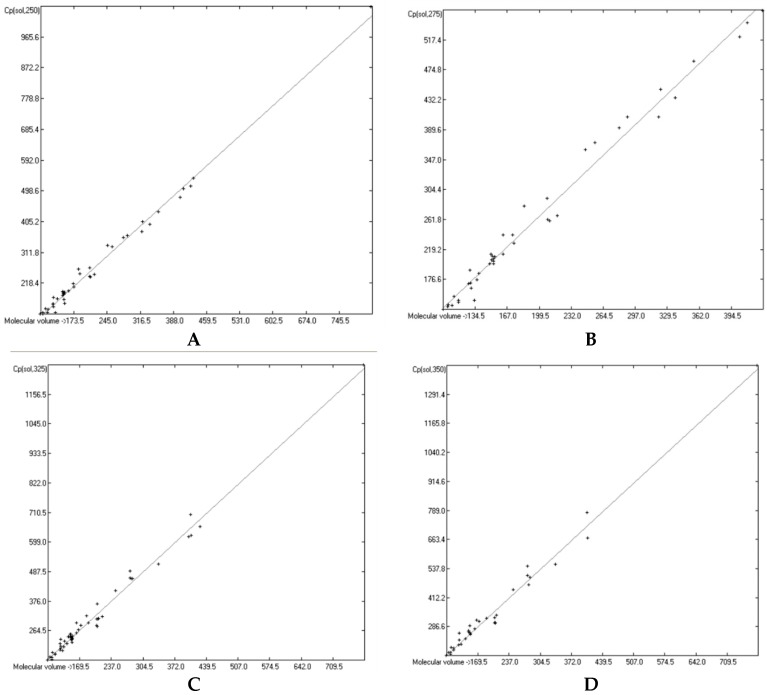
Correlation of molecular volumes with solid heat capacities of OH-carrying compounds at various temperatures. (**A**): 250 K. (**B**): 275 K. (**C**): 325 K. (**D**): 350 K.

**Table 1 molecules-24-01626-t001:** Comparison of “true” molecular volumes in A^3^ with literature references.

Molecule	Vm (Present Work)	Vm (ref. [41])	Vm (ref. [35])
Carbon dioxide	34.1	30.61	
Water	19.2	16.82	
Ammonia	22.7	21.91	
Methane	28.1	28.01	28.42
Ethane	44.9	44.63	45.38
Propane	61.5	61.39	62.37
Ethylene	39.6	40.25	39.64
Acetylene	34.8	36.15	38.35
Butadiyne	58.3	59.74	64.31
Benzene	83.4	85.39	80.28
Toluene	100	101.8	98.79
Biphenyl	153.3	157.1	152.2
Fluoroethane	49.8	47.45	49.17
Chloroethane	59.1	59.45	58.96
Bromobenzene	101.8	106.6	100.3
Iodobenzene	108	114.7	108.7
Cyclopropane	54.6	42.7/50	
Cyclohexane	100.5	99.1	100
Acetone	64.6	62.86	64.81
Methanol	36.9	34.89	36.04
Acetic acid	56.7	51.18	55.1

**Table 2 molecules-24-01626-t002:** Molecular volumes of popular cations and anions used in ionic liquids.

Cation	Vm (A^3^)	Anion	Vm (A^3^)
Methyltributylammonium	237.06	Nitrate	40.26
Pyrrolidinium	81.56	Acetate	54.06
1-Methyl-1-propylpyrrolidinium	145.36	Propionate	70.56
1-Butyl-1-methylpyrrolidinium	161.36	Butanoate	87.26
1-Methyl-1-pentylpyrrolidinium	178.76	Pentanoate	103.96
1-Hexyl-1-methylpyrrolidinium	195.46	Trifluoroacetate	68.66
1-Methyl-1-octylpyrrolidinium	228.86	Hydrogen sulfate	58.16
1-Butyl-1-methylpiperidinium	177.16	Methylsulfate	75.56
1,3-Dimethylimidazolium	97.76	Ethylsulfate	92.26
1-Ethyl-3-methylimidazolium	112.36	Octylsulfate	192.46
1-Butyl-3-methylimidazolium	146.2	Methanesulfonate	66.86
1-Hexyl-3-methylimidazolium	178.7	Trifluoromethanesulfonate	81.66
2,3-Dimethyl-1-hexylimidazolium	196.86	Toluenesulfonate	138.36
1-Octyl-3-methylimidazolium	203.91	Docusate	401.86
1-Decyl-3-methylimidazolium	248.36	Thiocyanate	54.76
1-Dodecyl-3-methylimidazolium	281.46	Tetrafluoroborate	46.46
1-Ethylpyridinium	114.16	Tetracyanoborate	100.26
1-Propylpyridinium	129.36	Dicyanoamide	57.26
1-Butylpyridinium	145.96	Tricyanomethide	76.26
1-Octylpyridinium	212.36	Dimethylphosphate	96.86
1-Ethyl-3-methylpyridinium	129.06	Diethylphosphate	130.36
1-Hexyl-3-methylpyridinium	196.76	Hexafluorophosphate	78.36
1-Ethyl-2-heptylpyridinium	227.96	Tris(pentafluoroethyl)trifluorophosphate	216.96
1-Ethyl-2-octylpyridinium	244.56	Bis(2,2,4-trimethylpentyl)phosphinate	304.86
1-Ethyl-2-nonylpyridinium	261.46	Bis(trifluoromethylsulfonyl)amide	156.46
4-Dimethylamino-1-hexylpyridinium	224.56	Saccharinate	138.06
Ethyl tributylphosphonium	270.16	Serinate	91.26
Tetrabutylphosphonium	302.96	L-Valinate	114.66
Tetradecyl trihexylphosphonium	570.96	L-Threoninate	105.16

**Table 3 molecules-24-01626-t003:** Experimental and Equation (2)-calculated Cp(liq.298) of 145 OH-free ionic liquids in J/mol/K.

Molecule Name	Cp(liq,298) exp	Cp(liq,298) calc	Deviation	Dev. in %
Pyrrolidinium nitrate	228.00	215.40	12.60	5.53
1-Ethyl-3-methylimidazolium bromide	264.80	256.10	8.70	3.29
3,3-Dinitroazetidinium nitrate	272.95	270.60	2.35	0.86
1-Propyl-3-methylimidazolium bromide	281.40	286.80	−5.40	−1.92
1-Ethyl-3-methylimidazolium thiocyanate	281.45	295.40	−13.95	−4.96
1-Ethyl-3-methylimidazolium acetate	321.90	296.60	25.30	7.86
1-Methyltetrahydrothiophenium dicyanamide	338.50	300.80	37.70	11.14
1-Ethyl-3-methylimidazolium tetrafluoroborate	305.00	302.10	2.90	0.95
1,3-Dimethylimidazolium methosulfate	341.00	305.20	35.80	10.50
1-Butyl-3-methylimidazolium chloride	317.00	305.90	11.10	3.50
1-Ethyl-3-methylimidazolium dicyanamide	314.64	314.90	−0.26	−0.08
1-Butyl-3-methylimidazolium bromide	317.00	317.90	−0.90	−0.28
1-Ethyl-3-methylimidazolium methanesulfonate	345.50	322.00	23.50	6.80
1-Ethyl-3-methylimidazolium trifluoroacetate	316.00	323.90	−7.90	−2.50
1-Ethyltetrahydrothiophenium dicyanamide	335.38	330.90	4.48	1.34
1-Butyl-3-methylimidazolium nitrate	353.50	333.30	20.20	5.71
1-Propylpyridinium tetrafluoroborate	363.00	335.40	27.60	7.60
1-Ethyl-3-methylimidazolium methylsulfate	341.00	338.70	2.30	0.67
1-Ethyl-3-methylimidazolium tricyanomethide	358.70	340.10	18.60	5.19
1-Ethyl-3-methylimidazolium hexafluorophosphate	343.60	341.30	2.30	0.67
1-Ethylpyridinium hexafluorophosphate	293.00	341.60	−48.60	−16.59
1-Butyl-3-methylimidazolium iodide	314.00	341.70	−27.70	−8.82
1-Benzyl-3-methylimidazolium chloride	339.40	343.80	−4.40	−1.30
1-Ethyl-3-methylimidazolium trifluoromethylsulfonate	362.80	349.60	13.20	3.64
1-Butyl-3-methylimidazolium thiocyanate	385.00	356.00	29.00	7.53
1-Butyl-3-methylimidazolium acetate	383.20	359.60	23.60	6.16
1-Butyl-3-methylimidazolium tetrafluoroborate	363.00	362.50	0.50	0.14
1-Methylazepanium methosulfate	332.08	364.90	−32.82	−9.88
1-Butylpyridinium tetrafluoroborate	383.80	365.60	18.20	4.74
1-Ethyl-3-methylimidazolium ethosulfate	378.00	366.70	11.30	2.99
1-Propylpyridinium hexafluorophosphate	328.70	371.90	−43.20	−13.14
1-Propyl-3-methylimidazolium hexafluorophosphate	374.40	372.00	2.40	0.64
1-Butyl-3-methylimidazolium dicyanoamide	365.00	373.00	−8.00	−2.19
1-Ethyl-3-methylimidazolium dimethylphosphate	411.78	375.20	36.58	8.88
1-Hexyl-3-methylimidazolium bromide	344.00	376.90	−32.90	−9.56
1-(3-Cyanopropyl)-pyridinium dicyanamide	422.00	379.40	42.60	10.09
1-Butyl-3-methylimidazolium trifluoroacetate	408.20	385.10	23.10	5.66
1-Butyl-2-methylimidazolium trifluoroacetate	407.90	388.90	19.00	4.66
1-Butyltetrahydrothiophenium dicyanamide	395.19	391.40	3.79	0.96
1-Butyl-2,3-dimethylimidazolium tetrafluoroborate	416.01	392.40	23.61	5.68
1-Butyl-3-methylpyridinium tetrafluoroborate	388.00	392.90	−4.90	−1.26
1-Butyl-4-methylpyridinium tetrafluoroborate	414.00	393.00	21.00	5.07
1-(3-Cyanopropyl)-2,3-dimethylimidazolium tetrafluoroborate	339.00	396.70	−57.70	−17.02
1-Ethyl-3-methylpyridinium ethylsulfate	389.00	397.10	−8.10	−2.08
1-Butyl-3-methylimidazolium methosulfate	416.00	397.30	18.70	4.50
1-Benzyl-3-methylimidazolium tetrafluoroborate	387.50	400.00	−12.50	−3.23
1-Butyl-3-methylimidazolium hexafluorophosphate	407.70	402.10	5.60	1.37
1-Butyl-1-methylpyrrolidinium dicyanamide	413.00	405.30	7.70	1.86
1-(3-Cyanopropyl)-2,3-dimethylimidazolium dicyanamide	444.00	408.90	35.10	7.91
1-Butyl-3-methylimidazolium trifluoromethylsulfonate	417.00	409.70	7.30	1.75
1-Hexyl-3-methylimidazolium tetrafluoroborate	416.00	421.50	−5.50	−1.32
1,2-Diethylpyridinium ethylsulfate	412.00	427.80	−15.80	−3.83
1-Butyl-2,3-dimethylimidazolium hexafluorophosphate	433.60	432.40	1.20	0.28
1-Pentyl-3-methylimidazolium hexafluorophosphate	437.40	432.90	4.50	1.03
1-Butyl-1-methylpyrrolidinium trifluoromethanesulfonate	435.00	437.40	−2.40	−0.55
1-Octyl-3-methylimidazolium bromide	392.00	438.00	−46.00	−11.73
1-Ethyl-3-methylimidazolium toluenesulfonate	484.20	451.10	33.10	6.84
1-Methyl-3-propylimidazolium 2-amino-4-carboxybutanoate	517.10	456.80	60.30	11.66
1-Hexyl-3-methylimidazolium hexafluorophosphate	424.00	461.50	−37.50	−8.84
1-Hexyl-3-methylimidazolium trifluoromethylsulfonate	502.30	470.00	32.30	6.43
Ethyl 1-ethylnicotinate ethosulfate	513.00	476.70	36.30	7.08
1-Butyl-1-methylpyrrolidinium tetracyanoborate	524.00	482.50	41.50	7.92
1-Octyl-3-methylimidazolium tetrafluoroborate	498.00	483.10	14.90	2.99
1-Ethyl-3-methylimidazolium bis(trifluoromethanesulfonyl) amide	500.00	483.20	16.80	3.36
1-Benzyl-3-methylimidazolium tetrafluoroethanesulfonate	502.80	485.10	17.70	3.52
1-Butyl-3-methylpyridinium tetracyanoborate	495.00	487.00	8.00	1.62
N-Ethylpyridinium bis(trifluoromethylsulfonyl)amide	502.15	487.20	14.95	2.98
1-Ethyl-3-methylimidazolium 2-(2-methoxyethoxy)ethylsulfate	526.00	492.90	33.10	6.29
1-Heptyl-3-methylimidazolium hexafluorophosphate	500.60	493.90	6.70	1.34
1-Ethyl-2,3-dimethylimidazolium bis(trifluoromethanesulfonyl) amide	493.00	508.50	−15.50	−3.14
1-Methyl-3-butylimidazolium saccharinate	565.66	509.30	56.36	9.96
1-Isopropyl-3-methylimidazolium bis(trifluoromethanesulfonyl) amide	529.90	512.70	17.20	3.25
1-Propyl-3-methylimidazolium bis(trifluoromethanesulfonyl) amide	534.90	513.10	21.80	4.08
1-Butyl-3-methylimidazolium toluenesulfonate	548.40	514.40	34.00	6.20
N-Ethyl-2-methylpyridinium bis(trifluoromethylsulfonyl)amide	534.50	515.50	19.00	3.55
1-Octyl-3-methylimidazolium hexafluorophosphate	536.10	524.50	11.60	2.16
1-Octyl-3-methylimidazolium trifluoromethylsulfonate	577.70	530.70	47.00	8.14
1-Cyclopropylmethyl-3-methylimidazolium bis(trifluoromethanesulfonyl) amide	539.10	532.60	6.50	1.21
Trimethyl butylammonium bis(trifluoromethylsulfonyl)amide	559.20	536.70	22.50	4.02
1,2-Diethylpyridinium bis(trifluoromethanesulfonyl) amide	566.10	538.80	27.30	4.82
1-Butyl-3-methylimidazolium bis(trifluoromethanesulfonyl) amide	536.00	542.00	−6.00	−1.12
1-sec-Butyl-3-methylimidazolium bis(trifluoromethanesulfonyl) amide	557.10	542.40	14.70	2.64
1-Isobutyl-3-methylimidazolium bis(trifluoromethanesulfonyl) amide	557.10	543.30	13.80	2.48
1-Methyl-1-propylpyrrolidinium bis(trifluoromethanesulfonyl) amide	554.00	543.30	10.70	1.93
1-Propyl-2,3-dimethylimidazolium bis(trifluoromethanesulfonyl) amide	554.50	543.60	10.90	1.97
N-Propyl-2-methylpyridinium bis(trifluoromethylsulfonyl)amide	557.96	545.80	12.16	2.18
N-Butylpyridinium bis(trifluoromethanesulfonyl) amide	566.52	546.70	19.82	3.50
N-Propyl-3-methylpyridinium bis(trifluoromethylsulfonyl)amide	517.00	547.20	−30.20	−5.84
1-Nonyl-3-methylimidazolium hexafluorophosphate	569.40	553.50	15.90	2.79
*N*-Octylisoquinolinium thiocyanate	522.00	557.90	−35.90	−6.88
*N*-Ethyl-4-dimethylaminopyridinium bis(trifluoromethanesulfonyl) amide	594.30	568.30	26.00	4.37
1-Pentyl-3-methylimidazolium bis(trifluoromethanesulfonyl) amide	595.60	571.30	24.30	4.08
1-Ethyl-2-propylpyridinium bis(trifluoromethanesulfonyl) amide	593.90	574.60	19.30	3.25
1-Isobutyl-1-methylpyrrolidinium bis(trifluoromethylsulfonyl)amide	582.20	576.20	6.00	1.03
1-Isobutyl-3-methylpyridinium bis(trifluoromethylsulfonyl)amide	579.00	576.70	2.30	0.40
N-Butyl-3-methylpyridinium bis(trifluoromethylsulfonyl)amide	578.10	577.30	0.80	0.14
1-Butyl-1-methylpyrrolidinium bis(trifluoromethylsulfonyl)amide	572.00	578.00	−6.00	−1.05
1-Butyl-3-cyanopyridinium bis(trifluoromethylsulfonyl)amide	586.00	579.50	6.50	1.11
1-Benzyl-3-methylimidazolium bis(trifluoromethylsulfonyl) amide	607.80	581.40	26.40	4.34
1-Decyl-3-methylimidazolium hexafluorophosphate	603.30	585.00	18.30	3.03
1-Cyclopentylmethyl-3-methylimidazolium bis(trifluoromethanesulfonyl) amide	560.60	586.30	−25.70	−4.58
Tri(butyl) methylphosphonium methylsulfate	617.80	592.70	25.10	4.06
1-Hexyl-3-methylimidazolium bis(trifluoromethanesulfonyl) amide	629.20	603.80	25.40	4.04
1-Butyl-1-methylpiperidinium bis(trifluoromethylsulfonyl)amide	607.50	606.10	1.40	0.23
1-Ethyl-2-butylpyridinium bis(trifluoromethanesulfonyl) amide	623.60	606.20	17.40	2.79
N-Hexylpyridinium bis(trifluoromethanesulfonyl) amide	612.00	606.90	5.10	0.83
1-Methyl-1-pentylpyrrolidinium bis(trifluoromethanesulfonyl) amide	622.60	608.40	14.20	2.28
1-Butyl-3-methylimidazolium octylsulfate	635.00	610.10	24.90	3.92
1-Cyclohexylmethyl-3-methylimidazolium bis(trifluoromethanesulfonyl) amide	617.10	615.50	1.60	0.26
N-Butyl-4-dimethylaminopyridinium bis(trifluoromethanesulfonyl) amide	657.71	626.70	31.01	4.71
1-Heptyl-3-methylimidazolium bis(trifluoromethanesulfonyl) amide	659.20	630.80	28.40	4.31
2,3-Dimethyl-1-Hexylimidazolium bis(trifluoromethanesulfonyl) amide	686.00	632.30	53.70	7.83
1-Ethyl-2-pentylpyridinium bis(trifluoromethanesulfonyl) amide	652.70	636.60	16.10	2.47
1-Hexyl-3-methylpyridinium bis(trifluoromethylsulfonyl)amide	624.00	637.70	−13.70	−2.20
1-Hexyl-1-methylpyrrolidinium bis(trifluoromethanesulfonyl) amide	655.10	638.80	16.30	2.49
1-Hexyl-4-cyanopyridinium bis(trifluoromethylsulfonyl)amide	633.00	639.60	−6.60	−1.04
1-Hexyl-3-cyanopyridinium bis(trifluoromethylsulfonyl)amide	658.00	639.70	18.30	2.78
1-Dodecyl-3-methylimidazolium hexafluorophosphate	666.30	646.50	19.80	2.97
1-Octyl-3-methylimidazolium bis(trifluoromethanesulfonyl) amide	654.00	664.00	−10.00	−1.53
1-Methyl-1-heptylpyrrolidinium bis(trifluoromethanesulfonyl) amide	685.10	664.80	20.30	2.96
1-Octylpyridinium bis(trifluoromethylsulfonyl)amide	686.00	665.90	20.10	2.93
1-Ethyl-2-hexylpyridinium bis(trifluoromethanesulfonyl) amide	685.50	667.10	18.40	2.68
1-Hexyl-3,5-dimethylpyridinium bis(trifluoromethylsulfonyl)amide	620.00	667.80	−47.80	−7.71
1-(3,4,5,6-Perfluorohexyl)-3-methylimidazolium-3-methylimidazolium bis(trifluoromethanesulfonyl) amide	725.00	678.60	46.40	6.40
4-Dimethylamino-1-hexylpyridinium bis(trifluoromethanesulfonyl) amide	731.00	687.20	43.80	5.99
N-Octyl-3-methylpyridinium bis(trifluoromethanesulfonyl) amide	669.00	697.10	−28.10	−4.20
1-Ethyl-2-heptylpyridinium bis(trifluoromethanesulfonyl) amide	717.50	697.40	20.10	2.80
1-Methyl-1-octylpyrrolidinium bis(trifluoromethanesulfonyl) amide	716.30	699.40	16.90	2.36
1-Octyl-3-cyanopyridinium bis(trifluoromethylsulfonyl)amide	709.00	700.10	8.90	1.26
1-Hexyl-3-methylimidazolium tris(pentafluoroethyl)trifluorophosphate	730.00	710.60	19.40	2.66
N-Hexyl-3-methyl-4-dimethylaminopyridinium bis(trifluoromethanesulfonyl) amide	725.00	710.90	14.10	1.94
Butyl 1-butylnicotinate bis(trifluoromethylsulfonyl)amide	707.00	713.40	−6.40	−0.91
Ethyl tri(butyl)phosphonium diethylphosphate	711.00	723.00	−12.00	−1.69
1-Decyl-3-methylimidazolium bis(trifluoromethanesulfonyl) amide	754.80	724.40	30.40	4.03
1-Ethyl-2-octylpyridinium bis(trifluoromethanesulfonyl) amide	749.40	725.00	24.40	3.26
1-Ethyl-2-nonylpyridinium bis(trifluoromethanesulfonyl) amide	778.50	751.40	27.10	3.48
Tetrabutylphosphonium L-valinate	747.00	753.40	−6.40	−0.86
1-Methyl-1-decylpyrrolidinium bis(trifluoromethanesulfonyl) amide	778.90	760.20	18.70	2.40
1-Dodecyl-3-methylimidazolium bis(trifluoromethanesulfonyl) amide	820.20	787.10	33.10	4.04
1-Ethyl-2-decylpyridinium bis(trifluoromethanesulfonyl) amide	811.20	788.60	22.60	2.79
1-Hexyl-2-propyl-3,5-diethylpyridinium bis(trifluoromethylsulfonyl)amide	766.00	809.50	−43.50	−5.68
Trihexyl tetradecyl phosphonium acetate	1078.20	1129.60	−51.40	−4.77
Tetradecyl trihexylphosphonium bis(trifluoromethylsulfonyl)amide	1298.80	1316.30	−17.50	−1.35
Trihexyltetradecylphosphonium tris(pentafluoroethyl)trifluorophosphate	1441.40	1435.80	5.60	0.39
Trihexyltetradecylphosphonium bis(2,2,4-trimethylpentyl)phosphinate	1635.00	1587.20	47.80	2.92
**Mean**			**9.30**	**1.60**
**Standard deviation**			**24.49**	**5.14**

**Table 4 molecules-24-01626-t004:** Experimental and Equation (2)-calculated Cp(liq.298) of 23 siloxanes in J/mol/K.

Molecule Name	Cp(liq,298) exp	Cp(liq,298) calc	Deviation	Dev. in %
Tetramethyoxysilane	240.50	239.40	1.10	0.46
Hexamethyldisiloxane	309.09	300.80	8.29	2.68
2,4,6,8-Tetramethylcyclotetrasiloxane	354.40	348.80	5.60	1.58
Hexamethylcyclotrisiloxane	360.00	350.90	9.10	2.53
Tetraethoxysilane	364.40	361.30	3.10	0.85
Octamethyltrisiloxane	420.64	416.20	4.44	1.06
Octamethylcyclotetrasiloxane	495.94	459.90	36.04	7.27
1,1,3,3-Tetraethyl-5,5-dimethylcyclotrisiloxane	502.90	471.10	31.80	6.32
Octamethyltetrasiloxane	509.60	474.50	35.10	6.89
Tetrapropoxysilane	460.10	482.70	−22.60	−4.91
1,1,3,3-Tetramethyl-1,3-diphenyldisiloxane	508.10	499.40	8.70	1.71
1,1,1,3,5,5,5-Heptamethyl-3-phenyltrisiloxane	519.60	517.10	2.50	0.48
Decamethyltetrasiloxane	538.77	529.60	9.17	1.70
Hexaethylcyclotrisiloxane	535.10	531.60	3.50	0.65
Decamethylcyclopentasiloxane	634.63	574.20	60.43	9.52
1,1,1,5,5,5-Hexamethyl-3,3-diphenyltrisiloxane	648.00	601.30	46.70	7.21
Tetrabutoxysilane	580.20	604.20	−24.00	−4.14
1,1-Diphenyl-3,3,5,5,7,7-hexamethylcyclotetrasiloxane	633.00	664.30	−31.30	−4.94
1,1,3,3-Tetraethyl-5,5-diphenylcylotrisiloxane	629.60	667.60	−38.00	−6.04
Octaethylcyclotetrasiloxane	746.00	695.50	50.50	6.77
Tetraheptoxysilane	909.40	968.50	−59.10	−6.50
Tetraoctoxysilane	1095.40	1089.90	5.50	0.50
Tetradecoxysilane	1373.50	1332.90	40.60	2.96
**Mean**			**8.14**	**1.50**
**Standard deviation**			**30.22**	**4.69**

**Table 5 molecules-24-01626-t005:** Experimental and Equation (2)-calculated Cp(liq.298) of 222 hydrocarbons in J/mol/K.

Molecule Name	Cp(liq,298) exp	Cp(liq,298) calc	Deviation	Dev. in %
Cyclopropane	81.20	94.00	−12.80	−15.76
Propylene	102.00	96.40	5.60	5.49
s-trans-1,3-Butadiene	123.65	116.90	6.75	5.46
1,2-Butadiene	123.00	117.30	5.70	4.63
1-Butyne	132.60	118.90	13.70	10.33
2-Butyne	124.30	119.10	5.20	4.18
Cyclobutane	106.30	120.40	−14.10	−13.26
Cyclopentadiene	115.30	122.50	−7.20	−6.24
cis-2-Butene	127.00	124.90	2.10	1.65
Isobutylene	121.30	126.00	−4.70	−3.87
trans-2-Butene	124.30	126.10	−1.80	−1.45
1-Butene	128.96	126.80	2.16	1.67
Cyclopentene	122.38	135.30	−12.92	−10.56
Butane	140.90	136.90	4.00	2.84
Methylenecyclobutane	133.60	138.20	−4.60	−3.44
Spiro[2.2]pentane	134.52	142.60	−8.08	−6.01
Isoprene	151.08	145.20	5.88	3.89
cis-1,3-Pentadiene	146.57	146.20	0.37	0.25
Benzene	136.80	146.30	−9.50	−6.94
1,4-Pentadiene	146.82	146.70	0.12	0.08
3-Methyl-1,2-butadiene	152.42	146.90	5.52	3.62
2,3-Pentadiene	152.34	147.00	5.34	3.51
trans-1,3-Pentadiene	149.33	147.00	2.33	1.56
1,2-Pentadiene	150.83	147.70	3.13	2.08
Cyclopentane	128.80	148.50	−19.70	−15.30
Trimethylethylene	152.80	154.50	−1.70	−1.11
Isopentene	157.30	155.70	1.60	1.02
cis-2-Pentene	151.80	156.10	−4.30	−2.83
1,3-Cyclohexadiene	141.30	156.30	−15.00	−10.62
trans-2-Pentene	157.00	156.40	0.60	0.38
3-Methyl-1-butene	156.10	156.80	−0.70	−0.45
1-Pentene	155.30	157.10	−1.80	−1.16
2,5-Norbornadiene	161.20	157.50	3.70	2.30
1,4-Cyclohexadiene	142.20	158.30	−16.10	−11.32
3-Methylcyclopentene	152.30	165.40	−13.10	−8.60
1-Methylcyclopentene	153.10	166.00	−12.90	−8.43
Isopentane	164.85	166.50	−1.65	−1.00
2,2-Dimethylpropane	153.10	166.90	−13.80	−9.01
Pentane	167.19	167.20	−0.01	−0.01
Cyclohexene	152.90	167.40	−14.50	−9.48
Quadricyclane	157.60	168.10	−10.50	−6.66
Methylbenzene	158.70	176.60	−17.90	−11.28
1,5-Hexadiene	133.10	177.20	−44.10	−33.13
Cyclohexane	158.10	177.30	−19.20	−12.14
Methylcyclopentane	158.70	178.40	−19.70	−12.41
Tetramethylethene	174.68	183.00	−8.32	−4.76
cis-2-Hexene	178.36	186.00	−7.64	−4.28
3,3-Dimethyl-1-butene	188.30	186.50	1.80	0.96
1-Hexene	183.30	187.50	−4.20	−2.29
Ethynylbenzene	180.10	188.20	−8.10	−4.50
1,3,5-Cycloheptatriene	162.76	189.30	−26.54	−16.31
Cyclooctatetraene	185.18	194.20	−9.02	−4.87
Norcarane	187.90	194.80	−6.90	−3.67
Styrene	201.90	195.50	6.40	3.17
Cycloheptene	171.70	195.50	−23.80	−13.86
Neohexane	189.67	196.50	−6.83	−3.60
2,3-Dimethylbutane	188.80	196.60	−7.80	−4.13
4-Methylcyclohexene	180.42	196.60	−16.18	−8.97
1-Ethylcyclopentene	188.30	196.90	−8.60	−4.57
Methylenecyclohexane	177.40	196.90	−19.50	−10.99
Ethylidenecyclopentane	181.20	197.00	−15.80	−8.72
3-Methylpentane	190.83	197.20	−6.37	−3.34
2-Methylpentane	193.96	197.40	−3.44	−1.77
Hexane	197.66	197.50	0.16	0.08
Cycloheptane	180.61	204.50	−23.89	−13.23
Ethylbenzene	185.78	205.70	−19.92	−10.72
Bicyclo[2.2.2]oct-2-ene	156.73	205.90	−49.17	−31.37
1,2-Dimethylbenzene	187.65	206.10	−18.45	−9.83
1,3-Dimethylbenzene	183.18	207.00	−23.82	−13.00
1,4-Dimethylbenzene	183.65	207.10	−23.45	−12.77
Indene	186.94	207.10	−20.16	−10.78
Methylcyclohexane	184.96	207.50	−22.54	−12.19
cis-1,2-Dimethylcyclopentane	190.80	207.80	−17.00	−8.91
Ethylcyclopentane	187.40	208.20	−20.80	−11.10
trans-1,3-Dimethylcyclopentane	190.80	208.50	−17.70	−9.28
1,1-Dimethylcyclopentane	187.40	208.70	−21.30	−11.37
1,5-Cyclooctadiene	208.10	209.10	−1.00	−0.48
Indane	190.25	215.80	−25.55	−13.43
1-Heptene	211.79	217.80	−6.01	−2.84
endo-2-Methylnorbornane	184.30	218.90	−34.60	−18.77
exo-2-Methylnorbornane	185.80	219.30	−33.50	−18.03
trans-Bicyclo[3.3.0]octane	180.30	220.30	−40.00	−22.19
cis-Bicyclo[3.3.0]octane	213.40	220.30	−6.90	−3.23
cis-Bicyclo[4,2,0]octane	258.60	221.20	37.40	14.46
cis- Cyclooctene	207.80	222.90	−15.10	−7.27
Naphthalene	196.06	224.70	−28.64	−14.61
alpha-Methylstyrene	202.10	224.70	−22.60	−11.18
Ethylidenecyclohexane	203.80	225.40	−21.60	−10.60
Triptane	213.51	225.90	−12.39	−5.80
3,3-Dimethylpentane	214.80	226.40	−11.60	−5.40
2,2-Dimethylpentane	221.12	226.80	−5.68	−2.57
3-Ethylpentane	219.58	227.00	−7.42	−3.38
2,3-Dimethylpentane	218.30	227.20	−8.90	−4.08
2,4-Dimethylpentane	224.22	227.40	−3.18	−1.42
Heptane	225.33	227.80	−2.47	−1.10
2-Methylhexane	222.92	227.90	−4.98	−2.23
Allylcyclopentane	202.90	228.40	−25.50	−12.57
Hemimellitene	216.44	233.90	−17.46	−8.07
1,2,4-Trimethylbenzene	213.11	234.50	−21.39	−10.04
Isocumene	214.72	235.10	−20.38	−9.49
Cumene	215.40	235.10	−19.70	−9.15
1,1-Dimethylcyclohexane	209.24	236.50	−27.26	−13.03
cis-1,4-Dimethylcyclohexane	212.09	236.70	−24.61	−11.60
trans-1,3-Dimethylcyclohexane	212.84	236.70	−23.86	−11.21
trans-1,2-Dimethylcyclohexane	209.41	237.30	−27.89	−13.32
1,3,5-Trimethylbenzene	207.85	237.50	−29.65	−14.27
Ethylcyclohexane	211.79	237.60	−25.81	−12.19
Cyclooctane	215.53	237.70	−22.17	−10.29
Diisobutylene	240.20	237.80	2.40	1.00
trans-1,4-Dimethylcyclohexane	210.25	238.00	−27.75	−13.20
cis-1,3-Dimethylcyclohexane	209.37	238.10	−28.73	−13.72
Propylcyclopentane	216.27	238.50	−22.23	−10.28
1-Octyne	242.14	240.40	1.74	0.72
4-Octyne	233.60	240.60	−7.00	−3.00
1,2,3,4-Tetrahydronaphthalene	217.44	246.50	−29.06	−13.36
trans-2-Octene	239.30	247.50	−8.20	−3.43
Caprylene	241.21	248.20	−6.99	−2.90
cis-Hydrindan	214.18	249.20	−35.02	−16.35
trans-Hydrindan	209.70	249.50	−39.80	−18.98
cis-Bicyclo[6.1.0]nonane	235.10	252.90	−17.80	−7.57
1-Methylnaphthalene	224.39	253.90	−29.51	−13.15
2-Methylnaphthalene	228.00	254.50	−26.50	−11.62
Isooctane	242.49	254.50	−12.01	−4.95
2,3,4-Trimethylpentane	247.32	255.10	−7.78	−3.15
2,3,3-Trimethylpentane	245.56	256.30	−10.74	−4.37
3,3-Dimethylhexane	246.60	256.60	−10.00	−4.06
2,2,3-Trimethylpentane	245.60	256.70	−11.10	−4.52
3-Methylheptane	250.20	257.00	−6.80	−2.72
2,5-Dimethylhexane	249.20	257.00	−7.80	−3.13
2-Methylheptane	252.00	257.10	−5.10	−2.02
Allylcyclohexane	233.50	257.40	−23.90	−10.24
4-Methylheptane	251.08	258.10	−7.02	−2.80
Octane	255.68	258.20	−2.52	−0.99
Tetrahydrodicyclopentadiene	236.50	261.10	−24.60	−10.40
Prehnitene	244.30	262.50	−18.20	−7.45
t-Butylbenzene	241.59	264.90	−23.31	−9.65
p-Cymene	242.30	264.90	−22.60	−9.33
Butylbenzene	243.50	265.50	−22.00	−9.03
1,2,4,5-Tetramethylbenzene	220.10	266.60	−46.50	−21.13
Propylcyclohexane	242.04	267.90	−25.86	−10.68
Butylcyclopentane	245.35	268.80	−23.45	−9.56
3-Carene	254.26	271.40	−17.14	−6.74
Biphenyl	259.54	273.40	−13.86	−5.34
4,7-Dimethylindane	241.50	275.30	−33.80	−14.00
1,1-Dimethylindane	249.40	275.50	−26.10	−10.47
4,6-Dimethylindane	240.90	275.60	−34.70	−14.40
Dihydropinene	251.26	277.10	−25.84	−10.28
trans-Decalin	229.17	278.60	−49.43	−21.57
1-Nonene	270.36	278.60	−8.24	−3.05
Bicyclopentyl	238.90	279.20	−40.30	−16.87
Tetraethylmethane	278.20	281.40	−3.20	−1.15
2,2,4,4-Tetramethylpentane	266.30	283.10	−16.80	−6.31
2,7-Dimethylnaphthalene	251.85	284.20	−32.35	−12.84
2,2,3,3-Tetramethylpentane	271.50	285.40	−13.90	−5.12
1-Methyl-4-isopropylcyclohexene	257.80	286.00	−28.20	−10.94
Nonane	284.43	288.40	−3.97	−1.40
Neopentylbenzene	271.90	295.10	−23.20	−8.53
t-Butylcyclohexane	264.80	295.60	−30.80	−11.63
Butylcyclohexane	271.04	298.30	−27.26	−10.06
Diphenylmethane	266.10	304.80	−38.70	−14.54
2-Phenyltoluene	275.77	304.80	−29.03	−10.53
1-Decene	300.83	308.90	−8.07	−2.68
5-Methylnonane	314.43	317.00	−2.57	−0.82
1-Ethyladamantane	258.45	317.70	−59.25	−22.93
4-Methylnonane	317.36	317.70	−0.34	−0.11
3-Methylnonane	308.99	317.70	−8.71	−2.82
Tolan	297.50	318.30	−20.80	−6.99
2-Methylnonane	313.30	318.40	−5.10	−1.63
1,3-Dimethyladamantane	258.31	318.80	−60.49	−23.42
Decane	316.32	318.80	−2.48	−0.78
cis,trans,trans-1,5,9-Cyclododecatriene	287.76	319.10	−31.34	−10.89
Hexamethylbenzene	370.70	322.00	48.70	13.14
1,1-Diphenylethylene	299.20	323.20	−24.00	−8.02
trans-Stilbene	343.10	323.40	19.70	5.74
1,2,3,4-Tetrahydrophenanthrene	278.26	324.00	−45.74	−16.44
1,1,4,7-Tetramethylindane	302.50	331.20	−28.70	−9.49
2-Ethylbiphenyl	302.73	331.50	−28.77	−9.50
1,2-Diphenylethane	320.10	333.10	−13.00	−4.06
3,3′-Bitolyl	295.45	333.20	−37.75	−12.78
2-Methyldiphenylmethane	296.58	334.00	−37.42	−12.62
2,2′-Dimethylbiphenyl	298.06	334.00	−35.94	−12.06
1,1-Diphenylethane	295.00	334.30	−39.30	−13.32
2,3-Dihydro-1,1,4,6-tetramethyl-1H-indene	299.60	335.60	−36.00	−12.02
Bicyclohexyl	283.00	337.80	−54.80	−19.36
1-Undecene	329.95	339.30	−9.35	−2.83
2-Methyldecane	341.21	348.80	−7.59	−2.22
Undecane	345.05	349.10	−4.05	−1.17
4-Isopropylbiphenyl	343.90	363.30	−19.40	−5.64
1-Dodecene	360.66	369.70	−9.04	−2.51
2,2,4,6,6-Pentamethylheptane	350.98	372.20	−21.22	−6.05
Dodecane	376.10	379.40	−3.30	−0.88
1,4-Di-t-butylbenzene	347.84	383.10	−35.26	−10.14
Heptylcyclohexane	363.20	389.40	−26.20	−7.21
4-t-Butylbiphenyl	383.60	391.40	−7.80	−2.03
1-Tridecene	391.80	400.00	−8.20	−2.09
m-Terphenyl	417.10	401.60	15.50	3.72
o-Terphenyl	369.05	402.10	−33.05	−8.96
Tridecane	407.10	410.40	−3.30	−0.81
Tetradecane	438.90	440.00	−1.10	−0.25
Decylcyclopentane	426.52	450.20	−23.68	−5.55
Pentadecane	469.90	470.30	−0.40	−0.09
Decylcyclohexane	455.60	479.70	−24.10	−5.29
2,2,4,4,6,8,8-Heptamethylnonane	458.80	488.00	−29.20	−6.36
Cetene	485.83	491.10	−5.27	−1.08
Cetane	501.61	500.60	1.01	0.20
Hexaethylcyclohexane	530.10	528.10	2.00	0.38
Heptadecane	534.34	531.00	3.34	0.63
Octadecane	564.40	561.30	3.10	0.55
Pristane	569.76	586.20	−16.44	−2.89
Nonadecane	604.00	591.60	12.40	2.05
Eicosane	663.60	622.90	40.70	6.13
1,1-Diphenyldodecane	593.70	636.60	−42.90	−7.23
1-Phenyl-1-cyclohexyldodecane	611.10	669.70	−58.60	−9.59
Docosane	698.00	683.70	14.30	2.05
4′-Heptyl-p-tercyclohexyl	752.70	709.60	43.10	5.73
4′-Heptyl-m-tercyclohexyl	668.60	709.60	−41.00	−6.13
Pentacosane	769.00	774.80	−5.80	−0.75
11-Cyclohexyleicosane	787.40	781.00	6.40	0.81
Heptacosane	828.40	835.50	−7.10	−0.86
Dodecahydrosqualene	886.36	915.70	−29.34	−3.31
11-Decylheneicosane	949.80	953.90	−4.10	−0.43
Unatriacontane	912.10	956.90	−44.80	−4.91
**Mean**			**−14.38**	**−6.34**
**Standard deviation**			**22.10**	**9.42**

**Table 6 molecules-24-01626-t006:** Molecular volumes Vm (in A^3^), experimental and Equation (3)-calculated Cp(liq.298) (in J/mol/K) of 164 monools and monoacids.

Molecule Name	Vm	Cp(liq,298) exp	Cp(liq,298) calc	Deviation	Dev. in %
Methanol	36.90	81.92	94.46	−12.54	−15.31
Formic acid	39.90	99.04	100.25	−1.21	−1.22
Ethanol	53.60	115.90	126.66	−10.76	−9.28
Acetic acid	56.70	123.10	132.64	−9.54	−7.75
Allyl alcohol	65.10	146.30	148.84	−2.54	−1.74
Colamine	65.60	137.70	149.80	−12.10	−8.79
Acrylic acid	67.40	144.20	153.27	−9.07	−6.29
1-Propanol	70.30	146.88	158.86	−11.98	−8.16
2-Propanol	70.30	154.43	158.86	−4.43	−2.87
Chloroacetic acid	70.40	179.90	159.06	20.84	11.58
Propanoic acid	73.30	158.60	164.65	−6.05	−3.81
Cyclobutanol	78.10	153.70	173.90	−20.20	−13.14
Methyl cellosolve	80.10	176.40	177.76	−1.36	−0.77
Hydroxymethyl acetate	82.40	214.00	182.19	31.81	14.86
Methacrylic acid	84.00	161.10	185.28	−24.18	−15.01
Dichloroacetic acid	84.50	188.00	186.24	1.76	0.94
2-Methyl-1-propanol	86.50	181.05	190.10	−9.05	−5.00
2-Butanol	86.50	196.67	190.10	6.57	3.34
2-Methyl-2-propanol	86.80	218.60	190.68	27.92	12.77
1-Butanol	87.00	177.16	191.06	−13.90	−7.85
2-Methylpropanoic acid	89.80	181.70	196.46	−14.76	−8.12
Furfuranol	89.80	204.01	196.46	7.55	3.70
Butanoic acid	89.90	177.70	196.66	−18.96	−10.67
Phenol	92.30	200.00	201.28	−1.28	−0.64
Cyclopentanol	93.40	185.40	203.40	−18.00	−9.71
2-Ethoxyethanol	96.90	210.30	210.15	0.15	0.07
Dimethylvinylcarbinol	97.80	208.40	211.89	−3.49	−1.67
Acetylacetone (enol form)	98.60	208.40	213.43	−5.03	−2.41
Cellosolve acetate	98.70	203.00	213.62	−10.62	−5.23
2-Aminoisobutanol	98.70	229.50	213.62	15.88	6.92
2-(Ethylamino)ethanol	99.40	227.00	214.97	12.03	5.30
Tetrahydrofurfuryl alcohol	102.40	190.00	220.76	−30.76	−16.19
2-Methyl-2-butanol	102.90	247.30	221.72	25.58	10.34
Isopentyl alcohol	103.00	209.60	221.91	−12.31	−5.87
3-Methyl-2-butanol	103.10	245.90	222.11	23.79	9.67
Neopentyl alcohol	103.50	244.30	222.88	21.42	8.77
1-Pentanol	103.60	208.14	223.07	−14.93	−7.17
3-Pentanol	103.80	239.70	223.46	16.24	6.78
3-Methylbutanoic acid	106.20	197.10	228.08	−30.98	−15.72
Pentanoic acid	106.60	210.00	228.86	−18.86	−8.98
o-Cresol	108.50	229.75	232.52	−2.77	−1.21
Benzyl alcohol	108.60	215.90	232.71	−16.81	−7.79
p-Cresol	108.80	221.03	233.10	−12.07	−5.46
m-Cresol	109.00	224.93	233.48	−8.55	−3.80
Cyclohexanol	109.50	213.40	234.45	−21.05	−9.86
2-Hydroxybenzaldehyde	109.90	222.00	235.22	−13.22	−5.95
2-Isopropoxyethanol	113.20	238.80	241.58	−2.78	−1.16
2-Propoxyethanol	113.60	241.60	242.35	−0.75	−0.31
cis-3-Hexen-1-ol	114.30	234.00	243.70	−9.70	−4.15
trans-3-Hexen-1-ol	114.60	237.20	244.28	−7.08	−2.98
1-Hexen-3-ol	114.60	269.30	244.28	25.02	9.29
2-(Isopropylamino)ethanol	115.50	258.82	246.02	12.80	4.95
Ethyl lactate	115.80	254.00	246.60	7.40	2.91
2-Methoxyphenol	118.10	240.00	251.03	−11.03	−4.60
Tetrahydro-2H-pyran-2-methanol	119.00	222.00	252.77	−30.77	−13.86
2-Ethyl-1-butanol	119.30	246.65	253.34	−6.69	−2.71
3-Methyl-3-pentanol	119.40	293.30	253.54	39.76	13.56
3,3-Dimethyl-1-butanol	119.60	236.08	253.92	−17.84	−7.56
4-Methyl-2-pentanol	119.70	272.34	254.12	18.22	6.69
3-Methyl-2-pentanol	119.70	275.89	254.12	21.77	7.89
3-Hexanol	119.90	269.27	254.50	14.77	5.49
2-Methyl-2-pentanol	120.00	289.03	254.69	34.34	11.88
2-Methyl-1-pentanol	120.10	248.40	254.89	−6.49	−2.61
4-Hydroxy-4-methyl-2-pentanone	120.50	221.30	255.66	−34.36	−15.53
1-Hexanol	120.60	242.70	255.85	−13.15	−5.42
2-Hexanol	120.60	256.31	255.85	0.46	0.18
n-Hexanoic acid	123.30	225.10	261.06	−35.96	−15.98
s-Phenethyl alcohol	124.20	257.45	262.79	−5.34	−2.07
Phenethyl alcohol	124.50	252.64	263.37	−10.73	−4.25
Cyclohexanemethanol	125.30	236.50	264.91	−28.41	−12.01
cis-2-Methylcyclohexanol	125.30	268.95	264.91	4.04	1.50
1-Methylcyclohexanol	125.30	279.05	264.91	14.14	5.07
Cycloheptanol	125.50	250.22	265.30	−15.08	−6.03
trans-2-Methylcyclohexanol	126.10	262.98	266.46	−3.48	−1.32
2,4-Dibromophenol	129.10	259.40	272.24	−12.84	−4.95
2-t-Butoxyethanol	129.40	273.45	272.82	0.63	0.23
Butylcellosolve	130.20	271.66	274.36	−2.70	−0.99
2-Diethylaminoethanol	130.80	281.20	275.52	5.68	2.02
2-Phenoxyethanol	134.80	294.63	283.23	11.40	3.87
1-Naphthol	135.40	284.50	284.39	0.11	0.04
Triethylmethanol	135.50	353.90	284.58	69.32	19.59
2,4-Dimethyl-3-pentanol	135.90	312.00	285.35	26.65	8.54
2,2-Dimethyl-3-pentanol	135.90	349.00	285.35	63.65	18.24
2-Methyl-2-hexanol	136.70	313.54	286.89	26.65	8.50
2-Methyl-3-hexanol	136.70	324.00	286.89	37.11	11.45
5-Methyl-2-hexanol	136.80	295.20	287.09	8.11	2.75
2-Heptanol	137.20	298.63	287.86	10.77	3.61
4-Heptanol	137.20	306.77	287.86	18.91	6.16
3-Heptanol	137.20	314.20	287.86	26.34	8.38
1-Heptanol	137.30	274.81	288.05	−13.24	−4.82
Enanthic acid	139.80	267.31	292.87	−25.56	−9.56
Hydrocinnamyl alcohol	141.10	280.74	295.38	−14.64	−5.21
Cyclohexaneethanol	142.50	266.00	298.08	−32.08	−12.06
2-Hydroxyethyl-2′,2′-dimethylpropionate	148.90	308.00	310.42	−2.42	−0.79
4-Ethyl-3-hexanol	152.20	361.30	316.78	44.52	12.32
Ethyl salicylate	152.70	283.07	317.75	−34.68	−12.25
4-Methyl-2-heptanol	152.70	312.50	317.75	−5.25	−1.68
3-Methyl-2-heptanol	152.80	297.50	317.94	−20.44	−6.87
4-Methyl-3-heptanol	152.80	309.20	317.94	−8.74	−2.83
3-Methyl-4-heptanol	152.80	355.80	317.94	37.86	10.64
2,5-Dimethyl-3-hexanol	152.90	339.40	318.13	21.27	6.27
2-Ethyl-1-hexanol	153.00	317.50	318.32	−0.82	−0.26
4-Methyl-4-heptanol	153.00	366.90	318.32	48.58	13.24
5-Methyl-2-heptanol	153.10	296.20	318.52	−22.32	−7.54
2-Methyl-4-heptanol	153.10	331.80	318.52	13.28	4.00
5-Methyl-1-heptanol	153.20	304.20	318.71	−14.51	−4.77
6-Methyl-3-heptanol	153.40	310.50	319.10	−8.60	−2.77
2-Methyl-2-heptanol	153.40	337.60	319.10	18.50	5.48
2-Methyl-1-heptanol	153.50	313.00	319.29	−6.29	−2.01
6-Methyl-2-heptanol	153.50	315.10	319.29	−4.19	−1.33
1-Octanol	153.70	312.10	319.67	−7.57	−2.43
4-Octanol	153.90	332.09	320.06	12.03	3.62
3-Octanol	153.90	338.50	320.06	18.44	5.45
2-Octanol	154.00	330.10	320.25	9.85	2.98
N-Methyl-2-hydroxyethylammonium propionate	154.40	328.00	321.02	6.98	2.13
n-Octanoic acid	156.60	304.00	325.27	−21.27	−7.00
Cyclohexanepropanol	159.20	293.00	330.28	−37.28	−12.72
4-Allyl-2-methoxyphenol	161.80	343.10	335.29	7.81	2.28
2-(2-(2-Methoxyethoxy)ethoxy)ethanol	166.10	357.05	343.58	13.47	3.77
1-Ethyl-3-methylimidazolium hydrogen sulfate	169.60	295.50	350.33	−54.83	−18.55
3,7-Dimethyl-6-octen-1-yn-3-ol	170.00	385.30	351.10	34.20	8.88
4-Nonanol	170.60	367.86	352.26	15.60	4.24
5-Nonanol	170.60	370.75	352.26	18.49	4.99
3-Nonanol	170.60	373.63	352.26	21.37	5.72
1-Nonanol	170.70	341.00	352.45	−11.45	−3.36
2-Nonanol	170.70	356.32	352.45	3.87	1.09
N-Methyl-2-hydroxyethylammonium butanoate	171.10	361.00	353.22	7.78	2.16
trans-Geraniol	172.60	346.10	356.12	−10.02	−2.90
Pelargonic acid	173.20	362.37	357.27	5.10	1.41
Diethyleneglycol monobutyl ether	173.30	354.89	357.47	−2.58	−0.73
Linalool	174.20	372.40	359.20	13.20	3.54
beta-Citronellol	179.30	357.90	369.04	−11.14	−3.11
1-Ethyl-3-methylimidazolium methylphosphonate	183.60	354.64	377.33	−22.69	−6.40
4-Propyl-4-heptanol	185.70	446.60	381.38	65.22	14.60
3,7-Dimethyl-1-octanol	186.10	367.21	382.15	−14.94	−4.07
n-Decyl alcohol	187.00	377.00	383.88	−6.88	−1.82
5-Decanol	187.30	405.77	384.46	21.31	5.25
N-Methyl-2-hydroxyethylammonium pentanoate	187.80	401.00	385.43	15.57	3.88
n-Decanoic acid	189.90	361.10	389.48	−28.38	−7.86
2-(2′-Hydroxyethoxy)ethyl pivalate	191.90	368.83	393.33	−24.50	−6.64
1-(2-Hydroxyethyl)-3-methyl-1H-imidazolium 2,2,2-trifluoroacetate	192.20	369.00	393.91	−24.91	−6.75
1-Undecanol	204.10	406.34	416.86	−10.52	−2.59
n-Undecanoic acid	206.50	415.10	421.48	−6.38	−1.54
1-Dodecanol	220.80	438.42	449.06	−10.64	−2.43
1H,1H-Perfluorooctan-1-ol	222.70	468.60	452.72	15.88	3.39
Lauric acid	223.20	404.28	453.68	−49.40	−12.22
Tributylmethanol	235.70	548.60	477.79	70.81	12.91
1-Tridecanol	237.50	476.00	481.26	−5.26	−1.11
1H,1H-Perfluorononan-1-ol	250.30	515.90	505.94	9.96	1.93
Pentaethylene glycol monomethyl ether	252.10	537.12	509.41	27.71	5.16
Myristyl alcohol	254.20	505.80	513.46	−7.66	−1.51
3,7,11-Trimethyl-1-dodecen-3-ol	262.20	574.50	528.88	45.62	7.94
1-Pentadecanol	270.90	535.10	545.66	−10.56	−1.97
1H,1H-Perfluorodecan-1-ol	274.50	563.90	552.60	11.30	2.00
1-Hexadecanol	287.60	523.80	577.86	−54.06	−10.32
1H,1H-Perfluoroundecan-1-ol	302.10	602.40	605.82	−3.42	−0.57
(9Z)-Octadecenoic acid	317.00	577.00	634.55	−57.55	−9.97
1H,1H-Perfluorododecan-1-ol	328.00	645.20	655.76	−10.56	−1.64
Methyltributylammonium serinate	331.10	635.00	661.74	−26.74	−4.21
3,7,11,15-Tetramethyl-1-hexadecyn-3-ol	340.80	712.50	680.44	32.06	4.50
Isophytol	344.10	729.70	686.80	42.90	5.88
Heptylpentaoxyethylene	352.40	722.90	702.81	20.09	2.78
1H,1H-Perfluorotetradecan-1-ol	379.80	725.30	755.64	−30.34	−4.18
Tetrabutylphosphonium L-serinate	389.50	748.00	774.34	−26.34	−3.52

**Table 7 molecules-24-01626-t007:** Molecular volumes Vm (in A^3^), experimental and Equation (4)-calculated Cp(liq.298) (in J/mol/K) of 36 polyols and polyacids.

Molecule Name	Vm	Cp(liq,298) exp	Cp(liq,298) calc	Deviation	Dev. in %
Ethan-1,2-diol	62.7	149.80	151.62	−1.82	−1.21
1,2-Propanediol	79.4	180.30	185.73	−5.43	−3.01
Propan-1,3-diol	79.4	175.78	185.73	−9.95	−5.66
DL-Lactic acid	81.8	210.50	190.63	19.87	9.44
Lactic acid	81.8	210.50	190.63	19.87	9.44
1,2,3-Propanetriol	88.2	218.90	203.70	15.20	6.94
2,3-Butyleneglycol	96.0	213.00	219.63	−6.63	−3.11
1,2-Butanediol	96.0	230.82	219.63	11.19	4.85
1,3-Butyleneglycol	96.1	218.89	219.83	−0.94	−0.43
Tetramethylene glycol	96.1	203.79	219.83	−16.04	−7.87
Butanedioic acid	100.9	238.60	229.64	8.96	3.76
Diethyleneglycol	105.8	243.90	239.64	4.26	1.75
1,5-Pentanediol	112.8	232.49	253.94	−21.45	−9.23
Pentanedioic acid	117.6	270.50	263.74	6.76	2.50
Hexyleneglycol	126.6	263.10	282.12	−19.02	−7.23
1,2-Hexanediol	129.4	293.10	287.84	5.26	1.79
Dipropylene glycol	138.8	322.10	307.04	15.06	4.68
1,7-Heptanediol	146.2	297.00	322.15	−25.15	−8.47
Triethyleneglycol	148.8	327.60	327.46	0.14	0.04
Heptanedioic acid	151.0	334.30	331.95	2.35	0.70
Octanedioic acid	167.2	366.20	365.03	1.17	0.32
Nonanedioic acid	184.4	398.10	400.16	−2.06	−0.52
Tetraglycol	191.8	428.90	415.27	13.63	3.18
Decanedioic acid	201.1	430.00	434.27	−4.27	−0.99
Tripropylene glycol	214.3	440.60	461.22	−20.62	−4.68
Undecanedioic acid	217.8	461.90	468.37	−6.47	−1.40
Dodecanedioic acid	234.5	493.80	502.48	−8.68	−1.76
Pentaglycol	234.8	515.50	503.09	12.41	2.41
1,13-Tridecanedioic acid	251.2	525.70	536.58	−10.88	−2.07
Tetrapropylene glycol	255.3	559.80	544.95	14.85	2.65
1,14-Tetradecanedioic acid	267.9	557.60	570.68	−13.08	−2.35
Hexaethylene glycol	277.8	620.10	590.90	29.20	4.71
Hexadecanedioic acid	301.3	621.40	638.89	−17.49	−2.81
Pentapropylene glycol	312.4	685.80	661.56	24.24	3.53
Ricinelaidic acid	325.8	646.00	688.93	−42.93	−6.65
Hexapropylene glycol	369.9	807.50	778.99	28.51	3.53

**Table 8 molecules-24-01626-t008:** Statistics of the correlations between molecular volumes and liquid heat capacities at 298.15K.

Molecules Class	N	Corr. coeff. R^2^	σ (J/mol/K)	Intercept	Slope
OH-free compounds	1102	0.9890	20.70	−5.3055	1.8183
All alcohols/acids	201	0.9724	23.24	20.9141	1.9676
Monoalcohols/-acids	164	0. 9685	22.91	23.3101	1.9282
Polyalcohols/-acids	36	0.9910	16.03	23.5782	2.0422

**Table 9 molecules-24-01626-t009:** Molecular volumes, experimental and calculated Cp(sol.298) of 123 monools and monoacids in J/mol/K.

Molecule Name	Vm	Cp(sol,298) exp	Cp(sol,298) calc	Deviation	Dev. in %
2-Methyl-2-propanol	86.80	146.11	119.41	26.70	18.27
Serine	91.30	135.60	126.18	9.42	6.94
Phenol	92.30	127.44	127.69	−0.25	−0.20
Dibromoacetic acid	92.80	124.70	128.44	−3.74	−3.00
2-Furoic acid	92.90	132.26	128.59	3.67	2.77
cis-3-Chloro-2-butenoic acid	96.20	140.20	133.56	6.64	4.74
4-Fluorophenol	97.20	144.60	135.06	9.54	6.59
trans-3-Chloro-2-butenoic acid	97.20	159.80	135.06	24.74	15.48
3-Thiophenecarboxylic acid	99.70	134.17	138.83	−4.66	−3.47
2-Thiophenecarboxylic acid	100.00	153.78	139.28	14.50	9.43
2-Pyrazinecarboxylic acid	102.50	143.50	143.04	0.46	0.32
5-Methylisoxazole-3-carboxylic acid	105.30	146.20	147.25	−1.05	−0.72
Trimethylacetic acid	106.20	177.80	148.61	29.19	16.42
Nicotinic acid	107.00	151.30	149.81	1.49	0.98
2-Pyridinealdoxime	108.00	151.93	151.32	0.61	0.40
o-Cresol	108.50	154.56	152.07	2.49	1.61
p-Cresol	108.80	150.20	152.52	−2.32	−1.55
Threonine	109.30	155.31	153.27	2.04	1.31
4-Amino-3-furazanecarboxamidoxime	109.70	180.40	153.88	26.52	14.70
Aspartic acid	110.50	155.18	155.08	0.10	0.06
Benzoic acid	111.10	147.07	155.98	−8.91	−6.06
4-Hydroxybenzaldehyde	111.50	149.90	156.59	−6.69	−4.46
Cyclohexanone oxime	114.90	199.38	161.70	37.68	18.90
4-Nitrophenol	115.40	143.90	162.45	−18.55	−12.89
3-Thiopheneacetic acid	115.70	167.76	162.91	4.85	2.89
2-Thiopheneacetic acid	116.50	159.23	164.11	−4.88	−3.06
Pentafluorophenol	116.80	201.30	164.56	36.74	18.25
3,5-Dimethylisoxazole-4-carboxylic acid	120.40	177.80	169.98	7.82	4.40
o-Anthranilic acid	121.00	165.30	170.88	−5.58	−3.38
m-Anthranilic acid	123.50	162.80	174.64	−11.84	−7.28
p-Anthranilic acid	123.60	177.80	174.80	3.00	1.69
4-Ethylphenol	124.50	206.90	176.15	30.75	14.86
2-Chlorobenzoic acid	124.70	163.20	176.45	−13.25	−8.12
2-Methylbenzoic acid	125.70	174.90	177.96	−3.06	−1.75
3-Chlorobenzoic acid	125.70	163.60	177.96	−14.36	−8.77
4-Chlorobenzoic acid	125.70	167.80	177.96	−10.16	−6.05
3-Methylbenzoic acid	127.80	163.60	181.12	−17.52	−10.71
4-Methylbenzoic acid	127.80	169.00	181.12	−12.12	−7.17
2-Bromobenzoic acid	128.60	154.00	182.32	−28.32	−18.39
Glutamic acid	128.70	175.06	182.47	−7.41	−4.23
Benzoylformic acid	129.60	192.90	183.82	9.08	4.70
3-Bromobenzoic acid	129.90	151.40	184.28	−32.88	−21.71
4-Bromobenzoic acid	129.90	151.40	184.28	−32.88	−21.71
2-Nitrobenzoic acid	130.70	191.60	185.48	6.12	3.19
8-Quinolinol	130.90	180.42	185.78	−5.36	−2.97
3-Phenylpropiolic acid	134.60	180.00	191.35	−11.35	−6.31
3-Nitrobenzoic acid	134.70	179.90	191.50	−11.60	−6.45
Benzofuran-2-carboxylic acid	134.70	184.80	191.50	−6.70	−3.63
4-Nitrobenzoic acid	134.80	180.30	191.65	−11.35	−6.30
2-Naphthol	135.30	172.80	192.40	−19.60	−11.34
1-Naphthol	135.40	166.90	192.55	−25.65	−15.37
Methylsalicylate	136.00	244.30	193.46	50.84	20.81
Tyramine	136.30	213.75	193.91	19.84	9.28
4-Hydroxy-3-methoxybenzaldehyde	137.10	187.60	195.11	−7.51	−4.00
4-Methoxybenzoic acid	137.20	207.50	195.26	12.24	5.90
2-Hydroxyacetanilide	137.60	182.40	195.86	−13.46	−7.38
Methyl 3-hydroxybenzoate	137.60	205.00	195.86	9.14	4.46
Methyl 4-hydroxybenzoate	137.70	214.41	196.01	18.40	8.58
trans-Cinnamic acid	138.60	197.50	197.37	0.13	0.07
Indole-2-carboxylic acid	138.90	178.40	197.82	−19.42	−10.89
N-Phenylglycine	139.40	177.40	198.57	−21.17	−11.94
Mesitol	140.20	189.60	199.78	−10.18	−5.37
3-Trifluoromethylbenzoic acid	142.20	223.60	202.79	20.81	9.31
Homocubane-4-carboxylic acid	143.50	207.00	204.74	2.26	1.09
Thionaphthene-2-carboxylic acid	143.60	179.00	204.89	−25.89	−14.47
2,5-Dibromobenzoic acid	147.00	223.00	210.01	12.99	5.82
2,4,6-Tribromophenol	147.20	172.00	210.31	−38.31	−22.27
Trimellitic anhydride	147.30	248.90	210.46	38.44	15.44
3,5-Dibromobenzoic acid	148.30	223.10	211.97	11.13	4.99
3-Aminocinnamic acid	150.00	227.60	214.53	13.07	5.74
2-Adamantanol	153.60	207.20	219.94	−12.74	−6.15
Ethyl vanillin	153.90	212.20	220.40	−8.20	−3.86
1-Adamantanol	154.00	196.70	220.55	−23.85	−12.12
Ethyl 3-hydroxybenzoate	154.40	217.22	221.15	−3.93	−1.81
Ethyl 4-hydroxybenzoate	154.40	233.93	221.15	12.78	5.46
D(+)-Carnitine	155.40	224.70	222.65	2.05	0.91
Hippuric acid	158.40	217.00	227.17	−10.17	−4.69
2,4,6-Trinitrophenol	160.00	239.70	229.58	10.12	4.22
2-Nitrocinnamic acid	160.10	240.60	229.73	10.87	4.52
2-Hydroxybiphenyl	161.50	227.61	231.83	−4.22	−1.86
Pentachlorophenol	161.80	201.96	232.28	−30.32	−15.02
3-Nitrocinnamic acid	161.90	240.20	232.43	7.77	3.23
4-Nitrocinnamic acid	161.90	238.10	232.43	5.67	2.38
Isoborneol	163.20	261.06	234.39	26.67	10.22
Tyrosine	164.40	216.44	236.20	−19.76	−9.13
2,3,5,6-Tetramethylbenzoic acid	172.80	231.70	248.84	−17.14	−7.40
2,3,4,5-Tetramethylbenzoic acid	174.40	249.50	251.25	−1.75	−0.70
Menthol	174.80	250.10	251.85	−1.75	−0.70
Diphenylmethanol	179.30	236.80	258.62	−21.82	−9.22
p-Methacryloyloxybenzoic acid	182.30	257.90	263.14	−5.24	−2.03
3-Cyano-4-methoxymethyl-5-nitro-6-methyl-(2-pyridone)	183.90	275.80	265.54	10.26	3.72
*N*-Methylephedrine	184.70	298.89	266.75	32.14	10.75
9-Hydroxy-1,4-anthraquinone	188.80	237.67	272.92	−35.25	−14.83
Pentamethylbenzoic acid	190.00	249.90	274.72	−24.82	−9.93
Benzoylglycylglycine	206.00	277.40	298.80	−21.40	−7.72
5-Hydroxyflavone	206.70	279.80	299.86	−20.06	−7.17
3-Hydroxyflavone	207.00	270.10	300.31	−30.21	−11.18
Ibuprofen	208.20	328.01	302.11	25.90	7.89
6-Hydroxyflavone	208.80	250.30	303.02	−52.72	−21.06
7-Hydroxyflavone	208.80	253.10	303.02	−49.92	−19.72
N-Methylephedrine hydrochloride	210.80	282.29	306.03	−23.74	−8.41
2-(6-Methoxynaphthalen-2-yl)propanoic acid	211.90	296.30	307.68	−11.38	−3.84
PMC	220.10	315.30	320.02	−4.72	−1.50
4-(Phenylmethoxy)benzeneacetic acid	224.60	340.00	326.80	13.20	3.88
Tri-t-butylmethanol	225.60	350.60	328.30	22.30	6.36
1-Tridecanol	237.50	378.00	346.21	31.79	8.41
Tridecanoic acid	239.90	387.60	349.82	37.78	9.75
Triphenylmethanol	248.60	318.80	362.91	−44.11	−13.84
Myristyl alcohol	254.20	388.00	371.34	16.66	4.29
Tetradecanoic acid	256.60	432.01	374.95	57.06	13.21
1-Pentadecanol	270.90	400.00	396.48	3.52	0.88
1-Pentadecanoic acid	273.30	443.28	400.09	43.19	9.74
*N*-(2-Hydroxy-4-methoxybenzylidene)-4-butylaniline	275.20	451.00	402.95	48.05	10.65
1-Hexadecanol	287.60	422.00	421.61	0.39	0.09
Palmitic acid	290.00	460.70	425.22	35.48	7.70
Margaric acid	306.70	475.70	450.35	25.35	5.33
Stearic acid	323.40	501.60	475.49	26.11	5.21
1-Nonadecanoic acid	340.10	525.40	500.62	24.78	4.72
Arachidic acid	356.80	545.10	525.75	19.35	3.55
N-Formylmethionine-leucyl-phenylalanine	403.70	567.60	596.33	−28.73	−5.06
Simvastatin	408.40	577.00	603.41	−26.41	−4.58
Pyrimethanil laurate	411.70	584.50	608.37	−23.87	−4.08
Methylprednisolone aceponate	426.90	604.80	631.25	−26.45	−4.37

**Table 10 molecules-24-01626-t010:** Molecular volumes Vm (in A^3^), experimental and calculated Cp(sol.298) (in J/mol/K) of 119 polyols and polyacids.

Molecule Name	Vm	Cp(sol,298) exp	Cp(sol,298) calc	Deviation	Dev. in %
Oxalic acid	67.90	91.00	90.02	0.98	1.07
Squaric acid	80.00	121.80	108.73	13.07	10.73
Malonic acid	81.70	127.63	111.36	16.27	12.75
Lactic acid	81.80	127.60	111.52	16.08	12.60
Maleic acid	94.10	135.60	130.54	5.06	3.73
Fumaric acid	96.00	141.80	133.47	8.33	5.87
Butanedioic acid	100.90	152.93	141.05	11.88	7.77
1,3-Dihydroxybenzene	101.00	135.53	141.20	−5.67	−4.19
1,4-Dihydroxybenzene	101.00	136.40	141.20	−4.80	−3.52
1,2-Dihydroxybenzene	101.10	144.30	141.36	2.94	2.04
Diethanolamine	108.30	137.00	152.49	−15.49	−11.31
2,2-Dimethyl-1,3-propanediol	112.30	183.18	158.68	24.50	13.38
Erythritol	113.40	161.90	160.38	1.52	0.94
Tartaric acid	114.40	184.50	161.92	22.58	12.24
TRIS	116.40	171.27	165.02	6.25	3.65
2,5-Dihydroxytoluene	117.30	174.90	166.41	8.49	4.86
2-Methylbutanedioic acid	117.50	199.60	166.72	32.88	16.47
Pentanedioic acid	117.60	186.90	166.87	20.03	10.72
cis-1,2-Cyclohexanediol	118.30	160.40	167.95	−7.55	−4.71
trans-1,2-Cyclohexanediol	118.40	163.20	168.11	−4.91	−3.01
Salicylic acid	118.50	160.90	168.26	−7.36	−4.58
4-Carboxyphenol	120.10	155.20	170.74	−15.54	−10.01
3,4-Dihydroxybenzaldehyde	120.30	205.90	171.05	34.85	16.93
3-Carboxyphenol	120.40	157.30	171.20	−13.90	−8.84
Pentaerythritol	127.60	188.40	182.33	6.07	3.22
Xylose	128.50	180.40	183.73	−3.33	−1.84
D-Ribose	128.50	183.20	183.73	−0.53	−0.29
Arabinose	128.90	184.10	184.34	−0.24	−0.13
Hexamethyleneglycol	129.50	190.00	185.27	4.73	2.49
Xylitol	130.00	207.00	186.05	20.95	10.12
Thiophenedicarboxylic acid	130.10	204.87	186.20	18.67	9.11
1,6-Anhydro-beta-D-glucopyranose	130.70	187.20	187.13	0.07	0.04
1,6-Hexanedioic acid	134.30	196.60	192.69	3.91	1.99
2,6-Pyridinedicarboxylic acid	134.60	166.31	193.16	−26.85	−16.14
Mandelic acid	136.00	199.20	195.32	3.88	1.95
Ethriol	137.10	213.80	197.02	16.78	7.85
Gallic acid	138.00	199.55	198.42	1.13	0.57
Phthalic acid	138.20	188.11	198.72	−10.61	−5.64
Isophthalic acid	138.30	201.70	198.88	2.82	1.40
Arabitol	138.90	201.35	199.81	1.54	0.77
Terephthalic acid	139.50	199.60	200.74	−1.14	−0.57
2-Deoxy-d-glucose	143.70	200.20	207.23	−7.03	−3.51
4,5-Dihydroxy-2-(dinitromethylene)imidazolidine	147.10	217.76	212.49	5.27	2.42
trans-4-Coumaric acid	147.60	193.00	213.26	−20.26	−10.50
Trimethylhydroquinone	148.30	217.60	214.34	3.26	1.50
Inositol	149.90	218.00	216.82	1.18	0.54
3-Chloromandelic acid	150.10	211.10	217.13	−6.03	−2.85
beta-D-Fructose	150.90	227.70	218.36	9.34	4.10
Heptanedioic acid	151.00	240.70	218.52	22.18	9.22
Vitamin B6	151.20	244.46	218.83	25.63	10.49
Citric acid	151.60	226.40	219.44	6.96	3.07
D-Psicose	151.60	221.30	219.44	1.86	0.84
D-Tagatose	152.40	228.00	220.68	7.32	3.21
alpha-D-Glucose	153.60	219.19	222.54	−3.35	−1.53
D-Galactose	153.80	213.70	222.85	−9.15	−4.28
D-Mannose	153.80	215.90	222.85	−6.95	−3.22
2,4-Dihydroxycinnamic acid	155.80	178.36	225.94	−47.58	−26.68
Caffeic acid	156.40	181.96	226.87	−44.91	−24.68
Dulcose	158.80	238.50	230.58	7.92	3.32
3-Nitrophthalic acid	159.80	247.02	232.12	14.90	6.03
4-Nitrophthalic acid	160.60	232.24	233.36	−1.12	−0.48
3,4-Dihydroxyhydrocinnamic acid	161.30	218.62	234.44	−15.82	−7.24
1,8-Octanediol	162.90	236.36	236.92	−0.56	−0.24
Sorbitol	163.50	239.00	237.84	1.16	0.48
Octanedioic acid	167.20	267.60	243.57	24.03	8.98
1-Methoxy-a-d-glucopyranoside	167.30	233.70	243.72	−10.02	−4.29
4,4′-Dihydroxybiphenyl	171.40	224.31	250.06	−25.75	−11.48
Ferulic acid	173.20	235.94	252.84	−16.90	−7.16
Dopa	173.30	260.60	253.00	7.60	2.92
1,4-Bis-(2-hydroxyethyl)piperazine	175.10	250.68	255.78	−5.10	−2.03
Glyceryl-2-benzoate	179.30	236.40	262.28	−25.88	−10.95
1,9-Nonanediol	179.60	256.74	262.74	−6.00	−2.34
1-Benzoylglycerol	179.80	238.90	263.05	−24.15	−10.11
2,6-Naphthalenedicarboxylic acid	182.80	230.30	267.69	−37.39	−16.23
Nonanedioic acid	184.40	294.50	270.16	24.34	8.26
Bisphenol F	188.50	246.30	276.50	−30.20	−12.26
N-Acetyl-d-glucosamine	189.30	276.90	277.74	−0.84	−0.30
1,10-Decanediol	196.30	279.26	288.56	−9.30	−3.33
Decanedioic acid	201.10	321.40	295.98	25.42	7.91
Bisphenol E	204.60	276.41	301.40	−24.99	−9.04
Bisphenol S	206.20	280.30	303.87	−23.57	−8.41
Arabinosylhypoxanthine	208.90	283.30	308.04	−24.74	−8.73
3,3′-Dihydroxy-4,4′-diaminodiphenylmethane	210.90	281.20	311.14	−29.94	−10.65
1,11-Undecanediol	213.00	297.79	314.38	−16.59	−5.57
Undecanedioic acid	217.80	348.30	321.81	26.49	7.61
Vidarabine	218.70	290.10	323.20	−33.10	−11.41
Bisphenol A	219.00	301.34	323.66	−22.32	−7.41
1-(2,2,3-Trimethyl-1,3-oxazolidin-5-yl)-butane-1,2,3,4-tetrol	219.90	331.46	325.05	6.41	1.93
Ellagic acid	226.60	328.85	335.41	−6.56	−2.00
1,12-Dodecanediol	229.70	330.23	340.21	−9.98	−3.02
Trolox	231.40	332.00	342.83	−10.83	−3.26
Dodecanedioic acid	234.50	375.20	347.63	27.57	7.35
1,13-Tridecanediol	246.40	366.88	366.03	0.85	0.23
3,5-Di-t-butylsalicylic acid	247.00	383.00	366.96	16.04	4.19
1,13-Tridecanedioic acid	251.20	402.10	373.45	28.65	7.13
1-Monocaprin	258.20	410.00	384.27	25.73	6.27
1,14-Tetradecanediol	263.10	379.61	391.85	−12.24	−3.22
1,14-Tetradecanedioic acid	267.90	429.00	399.27	29.73	6.93
Bisphenol AP	274.00	365.85	408.70	−42.85	−11.71
Cellobiose	277.50	436.10	414.12	21.98	5.04
Maltose	277.90	434.70	414.73	19.97	4.59
1,15-Pentadecanediol	279.80	377.45	417.67	−40.22	−10.66
Saccharose	281.50	424.30	420.30	4.00	0.94
Lactose	283.60	412.50	423.55	−11.05	−2.68
Glyceryl-2-laurate	290.30	436.40	433.91	2.49	0.57
1-Glyceryl laurate	291.30	447.70	435.45	12.25	2.74
1,16-Hexadecanediol	296.50	426.18	443.49	−17.31	−4.06
Hexadecanedioic acid	301.30	482.80	450.92	31.88	6.60
Vitamin B2	321.30	442.20	481.84	−39.64	−8.96
2-Monomyristin	323.50	506.30	485.24	21.06	4.16
Methylprednisolone	338.50	472.10	508.44	−36.34	−7.70
2-Monopalmitin	357.80	558.60	538.28	20.32	3.64
1-Palmitoylglycerol	358.10	566.90	538.74	28.16	4.97
2-Stearoylglycerol	391.10	610.40	589.77	20.63	3.38
1-Glyceryl stearate	391.70	610.40	590.70	19.70	3.23
Maltotriose	406.70	662.50	613.89	48.61	7.34
Tetraphenyltetrahydroxycyclotetrasiloxane	432.00	626.50	653.01	−26.51	−4.23
Cyclodextrin	776.30	1153.00	1185.38	−32.38	−2.81
Octaphenyltetrahydroxytricyclooctasiloxane	816.00	1273.00	1246.77	26.23	2.06

**Table 11 molecules-24-01626-t011:** Experimental and Equation (5)-calculated Cp(sol.298) of 9 metallocenes in J/mol/K.

Molecule Name	Cp(sol,298) exp	Cp(sol,298) calc	Deviation	Dev. in %
Azaferrocene	183.40	196.40	−13.00	−7.09
Nickelocene	202.39	200.70	1.69	0.84
Ferrocene	195.90	201.10	−5.20	−2.65
Manganocene	208.15	207.20	0.95	0.46
Acetylferrocene	246.00	253.40	−7.40	−3.01
1,1′-Diacetylferrocene	293.90	301.60	−7.70	−2.62
Benzylferrocene	312.60	320.70	−8.10	−2.59
Benzoylferrocene	313.60	323.80	−10.20	−3.25
1,1′-Dibenzoylferrocene	466.10	450.70	15.40	3.30
**Mean**			**−3.73**	**−1.85**
**Standar deviation**			**8.95**	**3.37**

**Table 12 molecules-24-01626-t012:** Experimental and Equation (5)-calculated Cp(sol.298) of 16 siloxanes in J/mol/K.

Molecule Name	Cp(sol,298) exp	Cp(sol,298) calc	Deviation	Dev. in %
Tetraisopropoxysilane	359.48	365.90	−6.42	−1.79
Tetrapropyl silicate	363.39	369.80	−6.41	−1.76
Tetra-2-butoxysilane	487.27	453.50	33.77	6.93
Tetra-2-methyl-1-propoxysilane	477.59	459.90	17.69	3.70
Tetrabutoxysilane	493.68	461.10	32.58	6.60
2,4,6-Trimethyl-2,4,6-triphenylcyclotrisiloxane	506.70	490.20	16.50	3.26
1,1,3,3-Tetraphenyl-1,3-dimethyldisiloxane	503.20	530.50	−27.30	−5.43
Tetra-3-methyl-1-butoxysilane	591.28	549.50	41.78	7.07
Tetramethyltetraphenylcyclotetrasiloxane	615.00	639.60	−24.60	−4.00
Hexaphenyldisiloxane	681.00	680.90	0.10	0.01
Hexaphenylcyclotrisiloxane	683.70	722.00	−38.30	−5.60
1,1,3,3,5,5-Hexaphenyl-7,7-dimethylcyclotetrasiloxane	815.50	801.20	14.30	1.75
Tetra-2-ethyl-1-hexoxysilane	870.88	816.50	54.38	6.24
Octaphenyltetracyclosiloxane	932.50	939.60	−7.10	−0.76
Octaphenylpentacyclosilsesquioxane	1129.00	1086.00	43.00	3.81
Octaphenyltetrahydroxytricyclooctasiloxane	1273.00	1246.80	26.20	2.06
**Mean**			**10.64**	**1.38**
**Standard deviation**			**28.70**	**4.40**

**Table 13 molecules-24-01626-t013:** Experimental and Equation (5)-calculated Cp(sol.298) of 107 hydrocarbons in J/mol/K.

Molecule Name	Cp(sol,298) exp	Cp(sol,298) calc	Deviation	Dev. in %
Benzene	97.90	116.90	−19.00	−19.41
2,4-Hexadiyne	133.57	128.60	4.97	3.72
Norbornene	129.90	138.10	−8.20	−6.31
Nortricyclene	129.00	140.80	−11.80	−9.15
Norbornane	151.00	149.60	1.40	0.93
Bicyclo[2.2.2]octane	157.69	171.00	−13.31	−8.44
Naphthalene	168.20	175.80	−7.60	−4.52
exo-Dicyclopentadiene	188.70	179.80	8.90	4.72
Bullvalene	190.38	189.30	1.08	0.57
Acenaphthylene	183.59	195.70	−12.11	−6.60
2-Methylnaphthalene	195.80	198.20	−2.40	−1.23
2,2,3,3-Tetramethylbutane	232.20	198.60	33.60	14.47
Adamantane	190.00	201.30	−11.30	−5.95
Acenaphthene	191.38	203.90	−12.52	−6.54
Camphene	235.13	205.70	29.43	12.52
1,2,4,5-Tetramethylbenzene	215.10	207.30	7.80	3.63
Biphenyl	198.17	212.40	−14.23	−7.18
1,8-Dimethylnaphthalene	242.80	215.10	27.70	11.41
Fluorene	203.13	219.20	−16.07	−7.91
Guajen	216.47	219.80	−3.33	−1.54
2,7-Dimethylnaphthalene	204.39	220.50	−16.11	−7.88
2,6-Dimethylnaphthalene	204.20	220.50	−16.30	−7.98
Pentamethylbenzene	210.50	225.80	−15.30	−7.27
Bicyclo[3.3.3]undecane	213.20	232.80	−19.60	−9.19
Phenanthrene	220.30	233.80	−13.50	−6.13
Anthracene	210.50	234.90	−24.40	−11.59
4-Phenyltoluene	238.74	235.10	3.64	1.52
Cyclohexylbenzene	263.17	235.90	27.27	10.36
Diphenylmethane	233.50	236.00	−2.50	−1.07
9,10-Dihydrophenanthrene	243.80	243.30	0.50	0.21
9,10-Dihydroanthracene	219.06	244.70	−25.64	−11.70
Tolan	225.90	246.10	−20.20	−8.94
Hexamethylbenzene	245.60	248.90	−3.30	−1.34
trans-Stilbene	235.00	250.00	−15.00	−6.38
Tetrahydroanthracene	247.02	251.10	−4.08	−1.65
4-Methylphenanthrene	262.80	251.90	10.90	4.15
Fluoranthene	230.25	253.50	−23.25	−10.10
Pyrene	229.70	256.40	−26.70	−11.62
1,2-Diphenylethane	253.76	257.30	−3.54	−1.40
4,4′-Dimethylbiphenyl	242.26	257.30	−15.04	−6.21
[2.2]Paracyclophane	252.70	261.40	−8.70	−3.44
Diamantane	223.40	262.10	−38.70	−17.32
[2,2]Metaparacyclophane	261.50	266.80	−5.30	−2.03
1,2,3,4,5,6,7,8-Octahydroanthracene	277.96	266.80	11.16	4.01
4,5,9,10-Tetrahydropyrene	257.12	273.20	−16.08	−6.25
1,2,3,6,7,8-Hexahydropyrene	255.54	280.50	−24.96	−9.77
[2,2]Metacyclophane	240.60	280.60	−40.00	−16.63
Triphenylene	259.20	290.70	−31.50	−12.15
Perhydrophenanthrene	289.50	291.60	−2.10	−0.73
Benz[a]anthracene	273.60	293.00	−19.40	−7.09
Tetracene	260.50	294.00	−33.50	−12.86
1,4-Di-t-butylbenzene	323.15	294.90	28.25	8.74
Tri-t-butylmethane	354.80	300.50	54.30	15.30
4-t-Butylbiphenyl	304.73	301.10	3.63	1.19
anti-trans-Truxane	275.30	303.50	−28.20	−10.24
p-Terphenyl	279.60	308.00	−28.40	−10.16
m-Terphenyl	281.00	308.70	−27.70	−9.86
o-Terphenyl	274.75	309.10	−34.35	−12.50
2,2′-Diindanyl	332.60	313.10	19.50	5.86
Perylene	274.90	313.20	−38.30	−13.93
Triptycene	283.10	319.90	−36.80	−13.00
1-Methyl-7-isopropylphenanthrene	294.60	321.90	−27.30	−9.27
[3,3]Paracyclophane	324.30	322.00	2.30	0.71
Tritane	308.80	332.60	−23.80	−7.71
Dicumyl	321.10	341.60	−20.50	−6.38
Triphenylethylene	309.20	347.30	−38.10	−12.32
1,1,1-Triphenylethane	316.70	351.80	−35.10	−11.08
1,1,2-Triphenylethane	319.70	352.80	−33.10	−10.35
1,2′-Dinaphthylmethane	314.60	352.90	−38.30	−12.17
Pentacene	311.50	353.20	−41.70	−13.39
Coronene	317.17	358.00	−40.83	−12.87
Cetane	441.80	383.20	58.60	13.26
4,4′-Di-t-butylbiphenyl	412.41	389.40	23.01	5.58
Hexacyclopropylethane	396.10	401.20	−5.10	−1.29
1,1′-Diphenyl-1,1′-bicyclopentyl	375.50	401.80	−26.30	−7.00
1,2,3-Triphenylbenzene	352.37	403.10	−50.73	−14.40
p-Quaterphenyl	363.50	404.70	−41.20	−11.33
1,3,5-Triphenylbenzene	361.00	404.70	−43.70	−12.11
m-Quaterphenyl	359.50	404.70	−45.20	−12.57
o-Quaterphenyl	359.10	406.40	−47.30	−13.17
Tetraphenylmethane	368.20	423.30	−55.10	−14.96
Octadecane	485.64	428.80	56.84	11.70
1,1,1,2-Tetraphenylethane	395.40	440.00	−44.60	−11.28
1,1′-Diphenyl-1,1′-bicyclohexyl	403.80	442.30	−38.50	−9.53
1,1,2,2-Tetraphenylethane	399.60	444.10	−44.50	−11.14
Tetraphenylethylene	388.70	445.00	−56.30	−14.48
Eicosane	479.90	475.10	4.80	1.00
o-Quinquephenyl	444.30	500.30	−56.00	−12.60
p-Quinquephenyl	455.50	500.60	−45.10	−9.90
m-Quinquephenyl	443.70	500.70	−57.00	−12.85
2,3-Dimethyl-2,3-bis(4-t-butylphenyl)butane	529.50	518.20	11.30	2.13
Docosane	563.60	520.80	42.80	7.59
2,11-Dicyclohexyldodecane	557.30	531.40	25.90	4.65
1,1-Dicyclohexyldodecane	562.60	532.00	30.60	5.44
Pentaphenylethane	473.60	534.50	−60.90	−12.86
3,4-Dimethyl-3,4-bis(4-t-butylphenyl)hexane	587.90	561.80	26.10	4.44
3,4-Diethyl-3,4-bis(4-t-butylphenyl)-hexane	631.30	600.70	30.60	4.85
Hexacosane	661.20	612.10	49.10	7.43
4,5-Diethyl-4,5-bis-(4-tert-butylphenyl)-octane	618.50	651.50	−33.00	−5.34
2,4,5,7-Tetramethyl-4,5-bis(4-t-butylphenyl)octane	683.90	651.90	32.00	4.68
Fullerene C70	664.21	685.90	−21.69	−3.27
4,5-Dipropyl-4,5-bis-(4-t-butylphenyl)octane	724.20	689.60	34.60	4.78
Dotriacontane	806.00	749.00	57.00	7.07
5,6-Dibutyl-5,6-bis(4-t-butylphenyl)decane	805.50	776.90	28.60	3.55
Hexatriacontane	881.00	840.40	40.60	4.61
Tetratetracontane	1049.60	1023.00	26.60	2.53
Pentacontane	1220.90	1159.90	61.00	5.00
**Mean**			**−8.83**	**−4.06**
**Standard deviation**			**38.20**	**9.60**

**Table 14 molecules-24-01626-t014:** Statistics of the correlations between molecular volumes and solid heat capacities at 298.15K.

Molecules Class	N	Corr. coeff. R^2^	σ (J/mol/K)	Intercept	Slope
OH-free compounds	555	0.9766	23.14	2.8899	1.3669
All alcohols/acids, sugars	242	0.9792	20.87	−14.7861	1.5364
Monoalcohols/-acids	123	0.9613	21.62	−11.2179	1.5050
Polyalcohols/-acids, sugars	119	0.9866	19.75	−14.9656	1.5462

**Table 15 molecules-24-01626-t015:** Statistics of liquid heat capacities of OH-free compounds at various temperatures.

Cp at Various K	N	Corr. coeff. R^2^	σ (J/mol/K)	Intercept	Slope
Cp(liq,250)	29	0.9588	16.08	−10.8602	1.7137
Cp(liq,275)	43	0.9765	13.58	−14.8530	1.7887
Cp(liq,298)	1100	0.9893	20.55	−6.8820	1.8302
Cp(liq,325)	94	0.9860	19.56	−12.6937	1.9223
Cp(liq,350)	87	0.9842	20.29	−4.3295	1.9458

**Table 16 molecules-24-01626-t016:** Statistics of liquid heat capacities of OH-carrying compounds at various temperatures.

Cp at Various K	N	Corr. coeff. R^2^	σ (J/mol/K)	Intercept	Slope
Cp(liq,250)	10	0.9702	11.04	7.4825	1.7070
Cp(liq,275)	36	0.8988	21.70	0.0897	1.9866
Cp(liq,298)	203	0.9696	24.79	17.7056	1.9911
Cp(liq,325)	59	0.9337	27.08	35.6984	2.0297
Cp(liq,350)	56	0.9605	21.51	39.5214	2.1639

**Table 17 molecules-24-01626-t017:** Statistics of solid heat capacities of OH-free compounds at various temperatures.

Cp at Various K	N	Corr. coeff. R^2^	σ (J/mol/K)	Intercept	Slope
Cp(sol,250)	81	0.9873	19.37	8.8062	1.1569
Cp(sol,275)	83	0.9883	17.71	5.5180	1.2659
Cp(sol,298)	559	0.9760	23.66	2.6919	1.3694
Cp(sol,325)	60	0.9721	23.48	−1.7976	1.4837
Cp(sol,350)	43	0.9711	25.59	−5.6718	1.5794

**Table 18 molecules-24-01626-t018:** Statistics of solid heat capacities of OH-carrying compounds at various temperatures.

Cp at Various K	N	Corr. coeff. R^2^	σ (J/mol/K)	Intercept	Slope
Cp(sol,250)	43	0.9916	15.50	−12.6827	1.2778
Cp(sol,275)	42	0.9846	15.10	−1.1741	1.3387
Cp(sol,298)	241	0.9743	20.77	−12.6463	1.5236
Cp(sol,325)	55	0.9859	21.94	−11.7895	1.6305
Cp(sol,350)	36	0.9846	28.61	−26.4116	1.8402

**Table 19 molecules-24-01626-t019:** Experimental and calculated liquid heat capacities of two compound samples at various temperatures in J/mol/K. (Calculations based on parameters of Table 15 and Table 16, respectively).

Example	T/K	Cp(liq,T) exp	Cp(liq,T) calc	Deviation	Dev in %
1-Phenylpyrazole [219]	275.00	222.35	223.04	−0.69	−0.31
	288.20	227.50	230.51	−3.01	−1.32
	298.15	231.60	236.50	−4.90	−2.12
	308.20	235.70	238.92	−3.22	−1.37
	325.00	236.19	242.97	−6.78	−2.87
1-Octanol [44]	250.00	251.00	269.85	−18.85	−7.51
	260.00	268.00	284.08	−16.08	−6.00
	275.00	285.00	305.43	−20.43	−7.17
	290.00	298.00	315.00	−17.00	−5.70
	298.15	312.10	319.6	−8.10	−2.60
	310.00	317.50	332.32	−14.82	−4.67
	325.00	336.00	347.66	−11.66	−3.47
	340.00	355.60	362.33	−6.73	−1.89
	350.00	368.30	372.11	−3.81	−1.04

**Table 20 molecules-24-01626-t020:** Experimental and calculated solid heat capacities of two compound samples at various temperatures in J/mol/K. (Calculations based on parameters of Table 17 and Table 18, respectively).

Example	T/K	Cp(sol,T) exp	Cp(sol,T) calc	Deviation	Dev in %
3,4-Dimethoxyphenylacetonitrile [195]	250.00	208.4	204.2	4.2	2.01
	265.00	218.2	213.3	4.9	2.26
	275.00	225.2	219.3	5.9	2.61
	280.00	228.8	222.5	6.3	2.77
	290.00	236.6	228.8	7.8	3.31
	298.15	243.5	233.9	9.6	3.94
	310.00	254.8	240.5	14.3	5.62
	320.00	265.6	246.0	19.6	7.37
	325.00	*	248.8		
2-Pyrazinecarboxylic acid [116]	250.00	125.2	118.3	6.9	5.52
	260.00	129.0	125.4	3.6	2.81
	275.00	134.3	136.0	−1.7	−1.30
	290.00	140.2	140.5	−0.3	−0.24
	298.15	143.2	143.0	0.2	0.14
	310.00	147.5	148.4	−0.9	−0.63
	325.00	153.1	155.3	−2.2	−1.46
	340.00	158.2	159.4	−1.2	−0.78
	350.00	161.7	162.2	−0.5	−0.31

* phase change.

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
