# Peer review of "Calculation of the Isobaric Heat Capacities of the Liquid and Solid Phase of Organic Compounds at and around 298.15 K Based on Their “True” Molecular Volume"

_molecules, 2019, doi:10.3390/molecules24081626_

Round 1

Reviewer 1 Report

This manuscript reports an extensive amount of systematic work with goal to develop a universally applicable method for the prediction of the isobaric heat capacities of the liquid and solid phase of molecules on the basis of their accurately approximated volume derived from their force-field optimized structure.  This study included an impressive number of compounds with known experimental heat capacities and it seems to be as complete as it could have ever been. The developed methodology was validated adequately, and it seems that it has potential to predict heat capacities of any species on purely theoretical basis.

This work has been performed with great care and the results are well exposed in the manuscript. I solely have small comments which could help the authors to make their article even clearer.

1.       It would be highly interesting to see a brief comment on the impact of the accuracy of the optimized structures (which is the basis for the whole subsequent investigation) and the resulting predicted heat capacities. Force-field based optimization may not be particularly accurate in selected cases. Could the impact of the chosen level of accuracy on the final results be estimated? Would the work benefit from more accurate determination of the molecular structures, or the gain would be negligible?

2.       Intermolecular interactions of polar molecules, and in the extreme case a strong interaction through hydrogen bonding (which is also a highly specific interaction), could have been accounted for in surprisingly smoothly. Again, similar to the above point, could a refinement of the structural optimization (e.g. consideration of self-association, or at least dimerization) possibly lead to meaningful improvements?

Author Response

Please find the reply attached as doc file.

Reviewer 2 Report

The specific value of the error, e.g. standard deviation or MAPD, only need to be shown in the results.  Readers or reviewers wont grasp the error magnitude by a bunch of numbers. So, youd better show which result is better than which result in the Abstract and Introduction.

Put the correlation figures, histogram figures and comparison tables in the Supplementary Materials.

In chapter 2, please give a flow chart to visually display the detailed steps of Geometry Optimization and Molecular Volume Calculation.  The strategy of Geometry Optimization and Molecular Volume Calculation is your original or existing programs?

In chapter 3, you classify free-OH/mono-OH/poly-OH and liquid/solid molecules, and give a linear regression result for each class respectively. Then, you study the effect of different temperature on heat capacity.  So, could heat capacity be expressed as Cp(T) = intercept(T) + slope(T) * Vm, if I want to know the heat capacity at any temperature?

Author Response

Please find the reply in the attached doc file.

Round 2

Reviewer 2 Report

The response is acceptable. But you'd better add your answers to the articles as possible, so that readers won't be confused by the similar questions.

Author Response

Answer 2 to Peer Reviewer 2

Many thanks for your comment. I assume that you are referring to the problem of comparing the results of the present paper with those of the cited publications. I have added therefore a few words at the end of the introductory section taking account of this point.

R. Naef